# Improved Delivery Performance of n-Butylidenephthalide-Polyethylene Glycol-Gold Nanoparticles Efficient for Enhanced Anti-Cancer Activity in Brain Tumor

**DOI:** 10.3390/cells11142172

**Published:** 2022-07-11

**Authors:** Ming-Tai Hsing, Hui-Ting Hsu, Chih-Hsuan Chang, Kai-Bo Chang, Chun-Yuan Cheng, Jae-Hwan Lee, Chien-Li Huang, Meng-Yin Yang, Yi-Chin Yang, Szu-Yuan Liu, Chun-Ming Yen, Shun-Fa Yang, Huey-Shan Hung

**Affiliations:** 1Institute of Medicine, Chung Shan Medical University, Taichung 40201, Taiwan; javafatmark@gmail.com (M.-T.H.); javawomanfanny@gmail.com (H.-T.H.); 2Department of Neurosurgery, Changhua Christian Hospital, Changhua 50006, Taiwan; 83998@cch.org.tw (C.-Y.C.); 182459@cch.org.tw (J.-H.L.); 182541@cch.org.tw (C.-L.H.); 3School of Medicine, Chung Shan Medical University, Taichung 40201, Taiwan; 4Department of Pathology, Changhua Christian Hospital, Changhua 50006, Taiwan; 5Graduate Institute of Biomedical Science, China Medical University, Taichung 40402, Taiwan; qsrgukp2010@gmail.com (C.-H.C.); kbwork2021@gmail.com (K.-B.C.); 6Department of Neurosurgery, Neurological Institute, Taichung Veterans General Hospital, Taichung 40705, Taiwan; yangmy04@gmail.com (M.-Y.Y.); jean1007@gmail.com (Y.-C.Y.); syliu@vghtc.gov.tw (S.-Y.L.); chunmingyen@gmail.com (C.-M.Y.); 7Graduate Institute of Medical Sciences, National Defense Medical Center, Taipei 11490, Taiwan; 8College of Nursing, Central Taiwan University of Science and Technology, Taichung 406053, Taiwan; 9College of Medicine, National Chung Hsing University, Taichung 40227, Taiwan; 10Department of Medical Research, Chung Shan Medical University Hospital, Taichung 40201, Taiwan; 11Translational Medicine Research, China Medical University Hospital, Taichung 40402, Taiwan

**Keywords:** n-butylidenephthalide, polyethylene glycol, gold nanoparticle, DBTRG human glioblastoma multiforme, anti-cancer capacity

## Abstract

n-butylidenephthalide (BP) has been verified as having the superior characteristic of cancer cell toxicity. Furthermore, gold (Au) nanoparticles are biocompatible materials, as well as effective carriers for delivering bio-active molecules for cancer therapeutics. In the present research, Au nanoparticles were first conjugated with polyethylene glycol (PEG), and then cross-linked with BP to obtain PEG-Au-BP nanodrugs. The physicochemical properties were characterized through ultraviolet-visible spectroscopy (UV-Vis), Fourier-transform infrared spectroscopy (FTIR), and dynamic light scattering (DLS) to confirm the combination of PEG, Au, and BP. In addition, both the size and structure of Au nanoparticles were observed through scanning electron microscopy (SEM) and transmission electron microscopy (TEM), where the size of Au corresponded to the results of DLS assay. Through in vitro assessments, non-transformed BAEC and DBTRG human glioma cells were treated with PEG-Au-BP drugs to investigate the tumor-cell selective cytotoxicity, cell uptake efficiency, and mechanism of endocytic routes. According to the results of MTT assay, PEG-Au-BP was able to significantly inhibit DBTRG brain cancer cell proliferation. Additionally, cell uptake efficiency and potential cellular transportation in both BAEC and DBTRG cell lines were observed to be significantly higher at 2 and 24 h. Moreover, the mechanisms of endocytosis, clathrin-mediated endocytosis, and cell autophagy were explored and determined to be favorable routes for BAEC and DBTRG cells to absorb PEG-Au-BP nanodrugs. Next, the cell progression and apoptosis of DBTRG cells after PEG-Au-BP treatment was investigated by flow cytometry. The results show that PEG-Au-BP could remarkably regulate the DBTRG cell cycle at the Sub-G1 phase, as well as induce more apoptotic cells. The expression of apoptotic-related proteins in DBTRG cells was determined through Western blotting assay. After treatment with PEG-Au-BP, the apoptotic cascade proteins p21, Bax, and Act-caspase-3 were all significantly expressed in DBTRG brain cancer cells. Through in vivo assessments, the tissue morphology and particle distribution in a mouse model were examined after a retro-orbital sinus injection containing PEG-Au-BP nanodrugs. The results demonstrate tissue integrity in the brain (forebrain, cerebellum, and midbrain), heart, liver, spleen, lung, and kidney, as they did not show significant destruction due to PEG-Au-BP treatment. Simultaneously, the extended retention period for PEG-Au-BP nanodrugs was discovered, particularly in brain tissue. The above findings identify PEG-Au-BP as a potential nanodrug for brain cancer therapies.

## 1. Introduction

Brain cancer involves a tumor, which usually occurs in the brain or central nervous system (CNS) [1] and can be classified as being either benign or malignant [2]. Benign brain cancer cells do not aggressively invade healthy cells and have a slow progression rate. According to the World Health Organization, malignant brain tumors are categorized into four grades: I, II, III, and IV, which are based upon their progression rates [3]. Grade I includes gangliocytoma, pilocytic astrocytoma, and craniopharyngioma [4]. Astrocytoma has been classified as a Grade II malignant brain tumor that will invade adjacent brain tissue, while also having the potential to relapse after therapeutic approaches, thereby possibly becoming a high-grade malignant tumor. Anaplastic astrocytoma is the Grade III malignant cancer, which also has a high tendency to relapse after clinical surgery. Finally, Grade IV involves a malignant tumor, which has a high cell density and replication rates with unusual cell morphology, such as glioblastoma multiforme (GBM). GBM can invade adjacent brain tissue over a large area, while also stimulating the abnormal proliferation of endothelial cells [5]. Similarly, clinical treatment for the four grades of malignant brain tumors involves radiation and chemotherapy; however, there are still high relapse rates after the therapies [6,7].

Conventional small molecule drug therapies can be categorized by several approaches, including both oral and injectable administration [8,9]. Conventional medications cause difficulties in drug administration for different pathologies, such as drug tolerance due to the change of pH values in a patient’s body [10]. Therefore, nanotechnology has been applied for decades in order to improve the shortcomings of conventional medications, as it explores the use of innovative carriers for purposes of drug delivery, including nanocapsules, microlipids, nanosuspension particles, and magnetic nanoparticles, as well as other options [11]. Nanodrug carriers can enhance delivery efficiency to specific sites, thus reducing degradation rates due to changes in pH values in the microenvironment. The drugs are encapsulated in the center of polymeric membranes to form nanodrug particles. The release rate of site-specific drugs during the delivery process can be facilitated through various types of polymers, such as natural and synthetic polymers [12]. These nano-scale strategies in clinical treatments can improve drug distribution and dispersibility to better reach superior therapeutic efficiency [13,14].

The blood-brain barrier (BBB) is a semipermeable 2 nm border composed of microvasculature vessels [15] and allows for small molecules, such as glucose, to penetrate it, as opposed to most other substances [16]. The function of the BBB is to protect the brain from the effects of hormones and neurotransmitters in order to better maintain a balanced microenvironment [17,18]. However, the BBB is an obstacle in the clinical treatment of encephalopathy. The 2 nm cellular gap junction causes conventional chemotherapy to have a low therapeutic efficiency. Therefore, nanodrug delivery systems have become innovative strategies, with the targeted-site delivery efficiency of nanocarriers being improved through various modifications without destroying the BBB [19,20].

Angelica is the most commonly used traditional Chinese medicine, due to its major therapeutic effects, including anti-atherosclerosis and anti-cancer abilities [21]. N-butylidenephthalide (denoted as BP in the current research) is a single purified compound extracted from Angelica using chloroform, with a molecular weight of 188.23 and the molecular formula of C_12_H_12_O_2_ [22]. Bioactive components produced by this natural substance have been verified as possessing superior capacities, such as anti-cancer, anti-angiogenic, anti-inflammatory, vasodilatory, and anti-atherosclerotic effects [23,24]. BP has been also demonstrated as owning the powerful capacity for cancer cell toxicity [25] and inducing stem cell differentiation at a low dose [26]. BP can cause DNA damage by interfering with cell cycle progression between the G0 and G1 phases [25], which in turn induces cancer cell apoptosis to achieve anti-cancer effects [27].

Polyethylene glycol (PEG) is a long-chain polymer composed of continuous ethylene oxide (CH_2_CH_2_O). The combination of PEG with drugs, proteins, nanoparticles, and microlipids has been verified to improve biocompatibility, biosafety, and therapeutic efficiency [28]. PEG is a hydrophilic substance with a unique excluded volume effect in aqueous solutions. Furthermore, PEG has the ability to resist protein adsorption, which does not easily cause an immune reaction, indicating its superior biocompatibility for drug delivery systems [29,30]. Gold nanoparticles (Au) have been widely used in proteins for drug delivery systems involving cancer treatments, owing to its ease of fabrication or functionalization with biomolecules, as well as its superior biocompatibility [31]. The literature has shown that Au nanoparticles are effective in controlling cancer cell growth [32]. Furthermore, Au can be applied as a radiosensitizing agent in radiation therapy. In addition, Au can facilitate DNA strand breaks through X-ray irradiation to achieve cancer cell proliferation [33]. The previous literature has demonstrated that the high expression of folic acid receptors or hyaluronic acid receptors in cancer cells could be specific targets for Au nanocarriers [34]. However, nanocarriers still have limitations, such as their permeability and drug retention efficiency. Therefore, modification with polymers in Au can enhance both targeting efficacy and cell uptake capacity [11]. The literature indicates that PEGylated Au nanoparticles had better anti-immune and anti-antigenic properties, which could prevent phagocytosis by the reticuloendothelial system (RES) and also avoid recognition and breakdown by proteolytic enzymes [35]. Au nanoparticles modified with PEG demonstrated both effective stability and penetration properties, allowing these carriers to circulate in the human body for longer periods of time and achieve the desired therapeutic efficiency. Previous research has also suggested that the combination of PEG and Au nanoparticles could enter the cancerous area, where a near-infrared light could be applied to enhance cancer treatment. Furthermore, due to the increased retention time of modified nanoparticles with anti-cancer drugs, the approach has been indicated as a novel clinical cancer treatment [36].

In this research, biocompatible PEG-Au nanocarriers cross-linked with BP were prepared. To investigate the potential of PEG-Au-BP in drug delivery, assessments of cell toxicity, cell uptake ability, endocytic routes, apoptosis of brain cancer cells, and biodistribution were all examined through in vivo and in vitro experiments. This investigation sought to explore a novel strategy while achieving a reduction in treatment costs for brain cancer patients.

## 2. Materials and Methods

### 2.1. Preparation of Polyethylene-Glycol-Gold Nanoparticles with n-Butylidenephthalide (PEG-Au-BP)

PEG solution was purchased from Sigma-Aldrich, USA (average mol wt = 200 kDa). An amount of 1 mL of PEG solution (500 µM) was diluted with 24 mL of deionized water to obtain 20 µM of PEG solution. Afterwards, 609.6 µL of PEG solution was mixed with 390.4 µL of 50 ppm physical gold nanoparticle solution (Gold NanoTech, Inc., Tapei, Taiwan) for 2 h to obtain the PEG-Au solution. N-Butylidenephthalide (BP) solution was purchased from Alfa Aesar, Ward Hill, NY, USA (10 mg/mL, 95%, mol weight: 188.23). To prepare the PEG-Au-BP solution, BP solution was first mixed with 1-ethyl-3-(3-dimethyl aminopropyl)-carbodiimide solution (EDC, 15 mM, ThermoFisher, Pittsfield, MA, USA) at volume ratio of 1:2 to interact for 2 h at room temperature. A total of 6 µL of the mixture was then combined with 250 µL of PEG-Au solution to react at 4 °C for 8 h. The PEG-Au-BP solution with 50 ppm of Au nanoparticles and 78 µg/mL of BP was obtained. Furthermore, to investigate the cellular metabolic process, 0.5 mg/mL of fluorescein isothiocyanate reagent (FITC-conjugated AffiniPure Goat Anti-Rabbit IgG, Jackson ImmunoResearch, West Grove, PA, USA) was conjugated with PEG-Au-BP through coupling with the amine groups at 4 °C for 8 h in the dark at a 50:1 volume ratio.

### 2.2. Material Characterizations

The as-prepared nanoparticles, namely Au, PEG-Au, PEG-Au-BP, and FITC-labeled PEG-Au-BP were characterized for their physical and chemical properties by using the same methodology performed in our previous research [37]. Dynamic light scattering (DLS) analysis was preceded by the use of a Malvern Zetasizer Nano ZS, and manipulated with a 532 nm light source at a 90° fixed scatter angle. One mL of each colloidal sample was added into a cuvette with a 1 cm optical path for the purpose of further measurements. The cumulant method was used to investigate intensity distribution values. Each as-prepared nanoparticle was investigated through a Zetasizer Nano ZS (Malvern Instruments, Malvern, UK) instrument with a 633 nm He-Ne laser at 25 °C. The UV-Vis spectrum was measured by a Helios Zeta spectrophotometer (Thermo Fisher, Waltham, MA, USA), where the wavelength ranged from 190 to 1100 nm, and 520 nm indicated typical absorption of Au nanoparticles. The data were further analyzed by Origin Pro 8 (Originlab Corporation, Northampton, MA, USA) software. Fourier-transform Infrared (FTIR) spectrometry (Shimadzu FTIR Model IRPrestige-21, Japan) was applied to obtain the detailed spectrum of the functional groups in the absorption frequency range of 400 to 4000 cm^−1^. The 1% (*w*/*w*) sample of each as-prepared material was first mixed with 100 mg of potassium bromide powder (Sigma, Burlington, MA, USA) and then pressed into a sheer slice. To improve the signal-to-noise percentage, an average of 32 scans for each material was measured. Furthermore, both a scanning electron microscope (SEM JEOL JEM-5200, JEOL Ltd., Akishima, Tokyo, Japan) and transmission electron microscope (TEM JEM 1010, JEOL Ltd., Akishima, Tokyo, Japan) were used to observe the Au nanoparticles. The test samples were prepared by adding 5 μL of nanoparticle suspension on a copper-coated TEM grid, before being dried out at room temperature. Afterwards, the voltage of TEM was set at 80 keV in order to observe the size and structure of the nanoparticles. Image Pro software (Media Cybernetics, Burlington, MA, USA) was used to analyze the size of the nanoparticles (n = 10).

### 2.3. Cell Culture of BAEC and DBTRG Cells

Non-transformed bovine aortic endothelial cells (BAEC) were cultured using low glucose Dulbecco’s modified Eagle’s medium (Invitrogen) containing 5.5 mM D-glucose (low glucose) and 4 mM L-glutamine. This was then supplemented with 10% FBS, 2.5 mg/L ECGS, 1% MEM non-essential amino acids, and pen-strep-neomycin. DBRTG human brain glioma cells (American Type Cell Culture, ATCC) were cultured with Roswell Park Memorial Institute (RPMI) 1640 medium supplemented with 10% FBS (GIBCO), 1% antibiotic solution (penicillin 5000 U/mL and streptomycin 5000 μg/mL), and 1% L-glutamine (200 mM). The above cell lines were cautiously incubated at 37 °C with 5% CO_2_ in a humidified atmosphere incubator.

### 2.4. Cytotoxicity Assay

First, the IC_50_ of BP was calculated. DBTRG cells (1 × 10^4^ per well) were treated with various concentrations (25, 50, 100, 200 µg/mL) of BP solution for 24 h. The cell growth was detected by MTT (Sigma-Aldrich, Burlington, MA, USA) assay. Based on the calculation, BP at the concentration of 50 µg/mL caused 50% of cell growth inhibition, as evaluated by using 95% confidence intervals (CI 95%) from non-linear regression when compared with the untreated control group. Therefore, 50 µg/mL of BP solution was chosen for further experiments.

Then, the cytotoxicity effect of each material (PEG, PEG-Au, PEG-Au-BP, and BP) on cell growth was also examined by MTT (Sigma-Aldrich, Burlington, MA, USA) assay. BAEC and DBTRG cells at a density of 1 × 10^4^ per well were cultivated in a 96-well culture plate at 37 °C with 5% CO_2_. After being incubated overnight for attachment, the cells were cultured with 1 mL of various materials containing medium for 24, 48, and 72 h. Next, the supernatant was removed after the incubation and 20 μL (0.5 mg/mL) of MTT solution was added into each well and incubated for 4 h at 37 °C. Afterwards, 200 μL of DMSO solution was added to terminate the reaction for 30 min at room temperature. The absorbance at 570 nm was used to determine the cell growth situation using a TECAN ULTRA microplate reader. The ratio of cell growth was calculated using the following formula: cell viability (%) = [(OD1 − OD0)/(OD2 − OD0)] ×100%. All the experiments were performed in triplicate.

### 2.5. Cellular Uptake and Endocytosis Investigation

To measure cell uptake ability, cells at the density of 1 × 10^4^ per well were seeded into a 24-well culture plate and cultured with FITC-labeled PEG-Au-BP (50 µg/mL) for 30 min, 2 h, and 24 h. The cells were washed using Phosphate-buffered Saline (PBS), and then fixed with 4% PFA for 15 min, 0.5% Triton X-100 (Sigma, Burlington, MA, USA) for permeability for 10 min, and F-actin staining (6 μM Rhodamine phalloidin, Sigma-Aldrich). Afterwards, the nucleus was stained with 4, 6-diamidino-2-phenylindole (DAPI) nuclear staining (50 μg/mL, Invitrogen, Waltham, MA, USA) for 10 min and washed twice with PBS. The fluorescence of each sample was observed through a fluorescence microscope (Zeiss Axio Imager A1) and quantified by Image J 5.0 software. Furthermore, the cellular uptake ability was also detected by fluorescein-positive cells through both a flow cytometer and fluorescence-activated cell sorting (FACS) software (Becton Dickinson, Canton, MA, USA).

Furthermore, to investigate the endocytic pathways, cells at the density of 1 × 10^4^ per well in 24-well culture plates were pre-treated with various endocytosis inhibitors for 1 h, including Cytochalasin D (Cyto-D, 1 μM), Chlorpromazine (CPZ, 2 μM), Bafilomycin A (Baf, 100 nM), and Methyl-βcyclodextrin (β-MCD, 2 mM). After being extensively washed, the cells were treated with FITC-labeled PEG-Au-BP and incubated for 30 min, 2 h, and 24 h. A total of 6 μM Rhodamine phalloidin was used for staining F-actin. Afterwards, the cells were collected and resuspended in PBS. The images were obtained with a fluorescence microscope (Zeiss Axio Imager A1) and quantified by Image J 5.0 software. The fluorescein-positive cells were also detected using FACS software (Becton Dickinson, Canton, MA, USA). All experiments were performed in triplicate.

### 2.6. LysoTracker Staining

The internal localization of FITC-labeled PEG-Au-BP (green) was observed by labeling cells with LysoTracker fluorescent probes (red). The cells at the density of 1 × 10^4^ per well were seeded in a 24-well plate (37 °C, pH 6.5) and incubated with nanoparticles for 30 min, 2 h, and 24 h. The cells were washed three times with a 4 °C PBS solution and stained with 50 nM LysoTracker Red (Invitrogen, Waltham, MA, USA) for 30 min in a 37 °C incubator. After staining, the cells were then washed thrice with PBS and fixed with 4% PFA before observing the intracellular delivery of nanoparticles through fluorescence microscopy and quantified by flow cytometry analysis. All experiments were performed in triplicate.

### 2.7. Cell Cycle Analysis

The DBTRG cells at the density of 2 × 10^5^ were cultured in 6-well plates and stood for cell attachment for a period of 48 h. Afterwards, the previous medium was removed and a new medium with various nanoparticles (PEG, PEG-Au, PEG-Au-BP, and BP) was added before incubating for 48 h. After incubation, the medium was removed and washed twice with PBS for further trypsinization. The cells were subsequently collected by a centrifuge (1500 rpm, 3 min) and suspended in 75% ethanol (1 mL, −20 °C) for 20 min. The dead cells were collected and mixed with 500 μL of PBS solution containing 10 μL of PI (1 mg/mL), Rnase A (5 mg/mL), and Triton X 100 (0.1%) for 30 min. The BD LSRFortessa™ Cell Analyzer (Becton Dickinson, Canton, MA, USA) was used to measure the cell cycle progress for 10,000 cells in each sample. The data were analyzed by BD FACSDiva™ software. All experiments were conducted in triplicate.

### 2.8. Annexin-V and PI Staining for Cell Apoptosis

DBTRG cells were first seeded at a density of 2 × 10^5^ cells per well in 6-well culture plates for overnight incubation at 37 °C prior to the experiment. Next, the cells were treated with various materials (PEG, PEG-Au, PEG-Au-BP, and BP) for a 48 h incubation period at 37 °C. Afterwards, the DBTRG cells were collected using 0.05% trypsin-EDTA, and treated with anti-Annexin-V (green color) and PI (red color) using an Annexin-V-FITC or Propidium Iodide (PI) apoptosis detection kit (BD Pharmingen). Test cells were stained with Annexin-V and PI (both 5 μL) and incubated in the dark for 15 min at room temperature, followed by 400 μL of Annexin-V binding buffer added into each sample. Annexin-V-positive cells were considered to be apoptotic cells, which were analyzed by a BD FACS Calibur flow cytometer (BD Biosciences, Canton, MA, USA). All experiments were conducted in triplicate.

### 2.9. Western Blot for Apoptotic Related Protein Expression

DBTRG cells at the density of 2 × 10^5^ cells per well were seeded in a 10 cm^2^ culture dish with various materials (PEG, PEG-Au, PEG-Au-BP, and BP) at 37 °C for 48 h. Cells were then collected using 0.05% trypsin-EDTA and washed twice with PBS solution, then added to lysis buffer (pH 7.5, Tris: 5 mL, NaCl: 3 mL, Nonidt P-40: 1 mL, 10% SDS: 1 mL, Sodium deoxycholate: 0.5 g) at 4 °C for 60 min. Afterwards, the cells were centrifuged at 4 °C at a rate of 13,000 rpm for 20 min in order to obtain the protein containing supernatant fluid.

The concentration of the protein was determined through a BCA Protein Assay Reagent Kit, which measured the absorption value at a 595 nm wavelength to determine the protein amount according to the standard curve. Following with the protocol (Bio-Rad Laboratories Inc., Hercules, CA, USA), the protein samples (25–30 μg) were subjected to sodium dodecyl sulfate polyacrylamide gel electrophoresis (SDS-PAGE) and then transferred onto Polyvinylidene Fluoride (PVDF) membranes (Immobilon P; EMD Millipore). After blocking in TBST with 5% milk powder for 1 h, the membranes were probed with primary antibodies (Bcl-2 (Santa Cruz, 1:1000 dilution), Bax (Santa Cruz, 1:1000 dilution), Cyclin D1 (Santa Cruz, 1:1000 dilution), p21 (Santa Cruz, 1:1000 dilution), and β-actin (Santa Cruz, Dallas, TX, USA, 1:5000 dilution)) at 4 °C overnight, as well as with the fluorescent secondary antibodies for 1 h at 25 °C. β-actin antibody was applied to ensure the uniformity of loading. The immunoblots were washed three times with TBST and incubated with HRP-conjugated goat, anti-rabbit, or anti-mouse IgG (1:2000 dilution) (Zhongshan Goldenbridge Biotechnology, Beijing, China) at room temperature for 1 h. The immunoblots were detected through an ECL kit (Beyotime Institute of Biotechnology, Jiangsu, China) and visualized after exposure to X-ray film. The protein expression levels were determined through Image J 5.0 software (Molecular Dynamics, Bethesda, MD, CA). The density of each protein band was normalized to β-actin. Image J software was used to visualize the protein bands and analyze the data for quantification. All experiments were conducted in triplicate.

### 2.10. In Vivo Tissue Distribution of PEG-Au-BP Nanoparticles

Animal experiments were conducted according to the National Institute of Health guidelines after approval by the Animal Care and Use Committee of China Medical University (IACUC-102-82-N). The weight of the male BALB/c nude mice used in the experiments was 15–20 g, with the age being 6–7 weeks. All mice were obtained from the National Laboratory Animal Center (Taiwan). FITC-labeled PEG-Au-BP was injected into the retro-orbital sinus of the mice, with the mice later sacrificed after either 24 or 48 h. The brain, heart, liver, spleen, lung, and kidney tissues were harvested and then fixed with 4% paraformaldehyde before being dehydrated and cautiously embedded in paraffin. The specimens were cryosectioned into sizes at 4 μm thickness for Hematoxylin and Eosin (H&E) staining (Sigma, Burlington, MA, USA) for histological measurements. To measure the biodistribution of PEG-Au-BP in each tissue, the samples were also observed through a fluorescence microscope. A green-colored-fluorescence indicated the presence of PEG-Au-BP nanoparticles.

### 2.11. Statistical Analysis

In this research, all the data are represented as the mean ± standard deviation (SD) taken from triplicate experiments in order to avoid uncertainty. Student’s *t*-tests were applied for statistical analysis of the differences between groups. A *p* value less than 0.05 was considered to be statistically significant.

## 3. Results

### 3.1. Characterization of Polyethylene-Glycol-Gold Nanoparticles with n-Butylidenephthalide (PEG-Au-BP)

Figure 1A indicates the brief procedure performed for preparation of polyethylene-glycol-gold nanoparticles cross-linked with n-butylidenephthalide (PEG-Au-BP). Afterwards, the physicochemical properties of the PEG-Au-BP nanoparticles were characterized through various methods. The absorption peak of each material was measured through a UV-Vis spectrophotometer. The peak at 520 nm demonstrates the presence of Au nanoparticles in PEG-Au and PEG-Au-BP (Figure 1B). The functional groups of various materials were observed and displayed as an FTIR spectrum. The specific peaks of PEG are seen at 3390 cm^−1^ (-OH stretching), 2880 cm^−1^ (C-H stretching), 1648 cm^−1^ (-OH blending), and 1108 cm^−1^ (C-O-C stretching) [38]. After PEG was conjugated with Au, the peak of 3390 cm^−1^ (-OH bond) shifted to 3397 cm^−1^, indicating Au nanoparticles were combined with PEG (Figure 1C). Furthermore, the peaks of 1653 cm^−1^ shifted to 1657 cm^−1^ (O=C-N, Amide bond), showing that PEG-Au was cross-linked with BP (Figure 1C).

Both size distribution intensity and size of various nanoparticles were investigated through a DLS analyzer. The percentage of intensity is demonstrated in Figure 2A and Appendix A. The Au nanoparticles were also observed by SEM (Figure 2B, left panel) and TEM (Figure 2B, right panel). Additionally, the diameter of Au, PEG-Au, PEG-Au-BP, and FITC-labeled PEG-Au-BP measured by DLS assay was 42.02 ± 8.95 nm, 177.5 ± 5.2 nm, 246.2 ± 7.0 nm, and 288.2 ± 4.2 nm, respectively (Figure 2C). The average size of Au nanoparticles was quantified as 44 nm through the SEM (Figure 2D), which corresponded to the results of DLS analysis. Moreover, the SEM images of PEG-Au and PEG-Au-BP nanoparticles are demonstrated in Figure 2E. Each nanoparticle was subjected to further experiments.

### 3.2. Cytotoxicity Assessments of PEG-Au-BP Nanodrugs on BAEC and DBTRG Cell Lines

BP has been verified as having better efficiency in inhibiting cancer cell proliferation. Therefore, the value of IC_50_ was calculated based on the concentration of BP, which caused 50% of cell growth inhibition. Appendix A indicates the cell viability of DBTRG cell lines after a 24 h incubation period using various concentrations of BP (25, 50, 100, and 200 μg/mL). After the calculation, the IC_50_ of BP in the DBTRG cell line was at the concentration of 50 μg/mL. Consequently, BP at the concentration of 50 μg/mL was chosen for further assessments.

The cytotoxicity of various nanomaterials was investigated through MTT assay after 24, 48, and 72 h of incubation. In the non-transformed BAEC cell line (Figure 3A), the cell viability (%) for 24 h of incubation is demonstrated (Control group (BAEC): 100%, PEG group: 89%, PEG-Au group: 95%, PEG-Au-BP group: 43%, and BP group: 36%). For a 48 h incubation period, the results are: control group (BAEC) at 100%, PEG group at 120%, PEG-Au group at 101%, PEG-Au-BP group at 59%, and BP group at 91%. Regarding 72 h of incubation, the data revealed the control group (BAEC) at 100%, PEG group at 106%, PEG-Au group at 97%, PEG-Au-BP group at 87%, and BP group at 85% (Figure 3A).

Furthermore, the cytotoxicity of brain cancer DBTRG cell line, as induced by various materials, was examined. The cell viability (%) results are displayed as follows: 24 h (control group (DBTRG): 100%, PEG group: 55%, PEG-Au group: 55%, PEG-Au-BP group: 21%, and BP group: 27%); 48 h (control group (DBTRG): 100%, PEG group: 85%, PEG-Au group: 84%, PEG-Au-BP group: 33%, and BP group: 37%); and 72 h (control group (DBTRG): 100%, PEG group: 68%, PEG-Au group: 72%, PEG-Au-BP group: 15%, and BP group: 24%) (Figure 3B). The above results demonstrate that PEG-Au-BP could significantly inhibit DBTRG brain cancer cell proliferation when compared to the PEG and PEG-Au groups.

### 3.3. Cell Uptake Ability between BAEC and DBTRG Cell Lines

Both BAEC and DBTRG cell lines were subjected to an investigation of cell uptake efficiency after being treated with FITC-labeled PEG-Au-BP nanodrugs. The green fluorescence of FITC-labeled PEG-Au-BP was detected through the immunofluorescence (IF) and fluorescence-activated cell sorting (FACS) methods. The uptake images of the BAEC (Figure 4A) and DBTRG cell lines (Figure 4D) were obtained at 30 min, 2 h, and 24 h. Afterwards, the uptake efficiency was quantified from the data taken from the IF and FACS methods. The results of BAEC in the IF method demonstrate that the PEG-Au-BP uptake amount remarkably increased to ~2.44-fold (*p* < 0.05) at 2 h and achieved ~3.56-fold (*p* < 0.01) at 24 h (Figure 4B) compared to the 30 min (1-fold) group. The uptake amount detected by the FACS method was ~1.72-fold (*p* < 0.001) at 2 h and ~1.75-fold (*p* < 0.001) at 24 h compared to the 30 min (1-fold) group (Figure 4C). Furthermore, PEG-Au-BP nanoparticles uptake by DBTRG brain cancer cells was also examined based on the IF and FACS methods. From the IF method, the uptake amount significantly increased to ~1.53-fold (*p* < 0.05) and ~2.13-fold (*p* < 0.05) at 2 and 24 h compared to the 30 min (1-fold) group, respectively (Figure 4E). The FACS results also demonstrate that the uptake efficiency was significantly higher at 2 and 24 h, which was ~2.11-fold (*p* < 0.001) and ~2.12-fold (*p* < 0.001), respectively (Figure 4F). The results indicate that PEG-Au-BP nanoparticles could be significantly taken up by BAEC and DBTRG brain cancer cells.

### 3.4. Investigation of Potential Cellular Transportation on BAEC and DBTRG Cell Lines

LysoTracker was applied to target lysosome in both BAEC and DBTRG brain cancer cells at various time points (30 min, 2 h, and 24 h). Lysosome is an important organelle involved with the uptake mechanism. Figure 5A demonstrates the images of BAEC treated with FITC-labeled PEG-Au-BP at each time point obtained from the IF method. Based on the results quantified by IF in Figure 5B, the FITC fluorescence intensity in BAEC significantly increased to ~1.44-fold (*p* < 0.01) at 2 h, then achieved ~1.58-fold (*p* < 0.01) at 24 h. Furthermore, a similar trend was also found in the DBTRG cell line. The images at 30 min, 2 h, and 24 h are demonstrated in Figure 5C. The uptake amount in DBTRG cells was quantified by the IF method. The FITC intensity in the DBTRG cells remarkably increased to ~1.22-fold (*p* < 0.05) and ~1.60-fold (*p* < 0.01) at 2 and 24 h, respectively, compared to the 30 min group (1-fold) (Figure 5D). Therefore, the above LysoTracker results illustrate that, after autophagy, PEG-Au-BP nanodrugs would not degrade in lysosome, which remained stable in both cell lines. This indicates that PEG-Au-BP nanoparticles have potential in drug delivery systems.

### 3.5. Measurements of Endocytic Routes between BAEC and DBTRG Cell Lines

During the cell uptake of foreign bodies, the endosome undergoes transportation and is metabolized by lysosomes. Thus, to investigate the mechanisms on endosomal effects, various lysosomal inhibiters (CPZ, β-MCD, Cyto-D, and Baf) were applied to identify the endocytotic pathways in both BAEC and DBTRG brain cancer cells. In Figure 6A, the images demonstrate the inhibition of uptake efficiency in BAEC at 30 min, 2 h, and 24 h. The quantified results based on the IF method indicate that the uptake ability of BAEC was significantly inhibited by CPZ and Baf. The results of the IF method (Figure 6B) demonstrate the following: CPZ at 30 min (~0.19-fold), 2 h (~0.55-fold), and 24 h (~0.36-fold); β-MCD at 30 min (~1.05-fold), 2 h (~0.83-fold), and 24 h (~0.74-fold); Cyto-D at 30 min (~1.07-fold), 2 h (~0.8-fold), and 24 h (~0.75-fold); and Baf at 30 min (~0.68-fold), 2 h (~0.32-fold), 24 h (~0.36-fold). The results of the FACS method (Figure 6C) displays CPZ at 30 min (~0.72-fold), 2 h (~0.91-fold), and 24 h (~0.80-fold); β-MCD at 30 min (~0.98-fold), 2 h (~0.99-fold), and 24 h (~0.92-fold); Cyto-D at 30 min (~1.02-fold), 2 h (~0.99-fold), and 24 h (~0.98-fold); and Baf at 30 min (~0.94-fold), 2 h (~0.96-fold), and 24 h (~0.81-fold). The illustration of endocytic routes for non-transformed BAEC is shown in Figure 6D.

Furthermore, the images showing DBTRG brain cancer cells treated with various inhibitors are also presented. Figure 6E indicates that the uptake ability of DBTRG cells were remarkably inhibited by CPZ and Baf. The quantified results based on the IF method (Figure 6F) are described as follows: CPZ at 30 min (~0.29-fold), 2 h (~0.28-fold), and 24 h (~0.34-fold); β-MCD at 30 min (~0.76-fold), 2 h (~0.73-fold), and 24 h (~0.82-fold); Cyto-D at 30 min (~0.67-fold), 2 h (~0.81-fold), and 24 h (~0.85-fold); and Baf at 30 min (~0.21-fold), 2 h (~0.29-fold), and 24 h (~0.34-fold). The FACS method (Figure 6G) displays CPZ at 30 min (~0.71-fold), 2 h (~0.87-fold), and 24 h (~0.88-fold); β-MCD at 30 min (~0.99-fold), 2 h (~1.01-fold), and 24 h (~0.95-fold); Cyto-D at 30 min (~1.03-fold), 2 h (~1.01-fold), and 24 h (~0.99-fold); and Baf at 30 min (~0.91-fold), 2 h (~0.97-fold), and 24 h (~0.89-fold). Figure 6H exhibits the endocytic mechanism for the DBTRG cell line. Based on these results, the endocytotic routes for absorbing PEG-Au-BP nanoparticles in both the BAEC and DBTRG brain cancer cell lines were verified as being significantly inhibited by both CPZ (clathrin-mediated endocytosis) and Baf (cell autophagy) lysosomal inhibitors.

### 3.6. Apoptotic Cell Death Regulation of PEG-Au-BP Nanodrugs on DBTRG Cell Lines

The cell cycle progression and apoptotic population of the DBTRG brain cancer cell line treated with various nanomaterials after 48 h was investigated by flow cytometry (Figure 7). The Sub-G1 phase was represented as cell apoptosis. Figure 7A demonstrates the cell cycle histograms of each treatment group. The data were further analyzed and quantified in Figure 7B. In the Sub-G1 phase, the percentage of apoptotic cells in the PEG-Au-BP and BP groups was 37.25% (*p* < 0.05) and 36.92% (*p* < 0.01), respectively, which was significantly higher than the other groups (control group: 4.83%, PEG group: 3.77%, and PEG-Au group: 2.61%). The DBTRG cells in the G0G1 phase illustrate the control group at 54.24%, PEG group at 53.27%, PEG-Au group at 52.27%, PEG-Au-BP group at 42.49%, and BP group at 44.92%. Furthermore, the S phase and G2M phase of the cells were also examined and determined to be: S phase (control group: 11.62%, PEG group: 15.42%, PEG-Au group: 24.78% (*p* < 0.05), PEG-Au-BP group: 7.45%, and BP group: 11.52%) and G2M phase (control group: 25.57%, PEG group: 27.81% (*p* < 0.05), PEG-Au group: 26.70%, PEG-Au-BP group: 15.29% (*p* < 0.05), and BP group: 15.87% (*p* < 0.001)) (Figure 7B). The above results indicate that PEG-Au-BP nanoparticles could significantly induce DBTRG brain cancer cells to apoptosis through regulating cell progression in the Sub-G1 phase.

Afterwards, Annexin-V/PI double staining assay was processed to detect the population of apoptotic cells for DBTRG brain cancer cells 48 h after each treatment. The histograms obtained from flow cytometry are demonstrated in Figure 7C. The viable cells in the PEG-Au-BP group (63.2%, *p* < 0.05) and BP group (61.7%, *p* < 0.001) were significantly lower than the other groups. The Annexin-V-positive cells were indicated as apoptotic cells, with both the PEG-Au-BP group (31.5%, *p* < 0.001) and BP group (19.3%, *p* < 0.001) being remarkably higher than the other groups. The population of dead cells was quantified, with both the PEG-Au-BP group (13.3%, *p* < 0.01) and BP group (17.2%, *p* < 0.05) again being significantly higher (Figure 7D).

Furthermore, the apoptotic-related protein expression in the DBTRG cell line induced by each nanoparticle was investigated by Western blotting assay after 48 h of incubation. The blotting images of various proteins are displayed in Figure 8A and the expression results are semi-quantified (Figure 8B). The expression of apoptotic resistance protein Cyclin D1 was significantly lower in the PEG-Au-BP group (~0.23-fold, *p* < 0.01) and BP group (~0.17-fold, *p* < 0.001), followed by the PEG group (~0.66-fold, *p* < 0.01), PEG-Au group (~0.75-fold, *p* < 0.01), and control group (1-fold). A similar condition was found in the expression of anti-apoptotic protein Bcl-2, with the results arranged as follows: PEG-Au-BP group (~0.38-fold, *p* < 0.001), BP group (~1.07-fold, *p* < 0.01), PEG group (~1.24-fold), PEG-Au group (~1.36-fold, *p* < 0.001), and control group (1-fold). However, the expression of apoptotic cascade proteins p21, Bax, and active caspase-3 was verified as being significantly induced in DBTRG brain cancer cells by PEG-Au-BP treatment, with the quantification results demonstrating p21 at ~4.29-fold (*p* < 0.01), Bax at ~3.73-fold (*p* < 0.001), and act-caspase-3 at ~7.19-fold, (*p* < 0.001). The above results determine that PEG-Au-BP could significantly induce the expression of apoptotic proteins in DBTRG cells to cause cell apoptosis.

### 3.7. Retention Efficiency of PEG-Au-BP Nanodrugs through In Vivo Assessment

To investigate the effects of PEG-Au-BP nanomedicine through an in vivo *assessment*, the PEG-Au-BP nanoparticles were conjugated with FITC and then injected into the retro-orbital sinuses of mice by orbital injection (Figure 9A). The tissue morphology and particle biodistribution were examined in the brain (forebrain, cerebellum, and midbrain), heart, liver, spleen, lung, and kidney 12 and 24 h after injection. The images of H&E staining in Figure 9B and demonstrate that PEG-Au-BP nanodrugs would not destroy the tissues 12 and 24 h after injection. Afterwards, the biodistribution of FITC-labeled PEG-Au-BP nanoparticles was also observed at 12 and 24 h with a fluorescence microscope. The fluorescence images of various tissues are shown in Figure 9C and Appendix A. The images indicate that the FITC-loaded PEG-Au-BP could be observed in most organs or tissues after orbital injection. Furthermore, the fluorescence intensity of FITC-labeled PEG-Au-BP in each tissue were also quantified and are displayed in Figure 9D and Appendix A. The average fluorescence intensity results at 12 and 24 h are demonstrated as follows: 12 h in the forebrain (32), cerebellum (609.67), midbrain (1381.67), heart (3319), liver (3130.67), spleen (5987.33), lung (4281.67), and kidney (3465.33); and 24 h in the forebrain (27), cerebellum (2882.67), midbrain (4608.33), heart (1812.67), liver (8290.33), spleen (7014), lung (6323.67), and kidney (9738.33). Additionally, we also explored that PEG-Au-BP nanodrug could be observed in stomach and spinal cord tissue (Appendix A). The above evidence indicates that PEG-Au carrying BP nanomedicine could act as a highly secure and superior retention efficiency treatment for the DBTRG human glioma cell line.

Figure 10 outlines a brief summary for the efficiency of PEG-Au-BP nanodrugs, which could significantly inhibit the viability of DBTRG human glioma cells as well as remarkably induce the expression of apoptotic-related protein (tumor-cell selective cytotoxicity). Furthermore, the evidence from the in vivo orbital injection illustrated that PEG-Au-BP nanoparticles could be a highly secure drug for the brain (forebrain, cerebellum, and midbrain), heart, liver, spleen, lung, and kidney, which could help maintain tissue integrity. PEG-Au-BP nanoparticles were also found to have an extended retention period in most organs and tissues, thus supporting PEG-Au-BP as a potential drug for anti-brain-cancer treatment.

## 4. Discussion

Currently, clinical treatments for cancer diseases mainly involve conventional radiation therapy, tumor resection surgery, and chemotherapy [39]. Radiation therapy may cause damage to the surrounding healthy tissue, even though it can effectively kill cancer cells [40]. In addition, chemotherapy is the treatment used when injecting the drug into patients, where it is retained in the body through blood circulation. Unfortunately, the non-specific targeting of chemotherapeutic drugs may cause toxicity in both normal and cancerous cells, and the drugs may not remain in the bloodstream for a long enough retention period necessary for them to reach therapeutic efficacy [41]. Furthermore, the cancer cells may offer resistance to chemotherapy due to the hypoxia and acidosis in the microenvironment of the tumors [42]. Therefore, drug carriers, which incorporate nanotechnology, provide various innovative strategies for brain cancer therapy. The drug carrier systems can specifically bind to the receptors of tumor cells to strengthen the efficacy of cytotoxicity [43].

Nanocarriers can be coated with biodegradable and biocompatible polymers, such as hyaluronic acid and polyethylene glycol, controlling the degradation rate for drug release and increasing the retention time of nanocarriers in the blood stream [44]. Nanocarriers can also encapsulate various drugs in polymeric substrates, which can be gradually hydrolyzed to release the drugs slowly [45]. Polyethylene glycols are polymeric biomaterials possessing the characteristics of biocompatibility, degradability, and bio-absorbability, and tend to be water soluble. Gold nanoparticles (Au) are the most promising candidate to be the inorganic core for the drug delivery system, owing to their superior biocompatibility and ease in functionalizing with bio-active molecules [46,47]. Previous research has elucidated that Au fabricated with degradable polymers, such as collagen and polyethylene glycol (PEG) could form biocompatible nanodrug systems [48,49]. According to the previous literature, the stability of Au nanoparticles could be modified by PEG with hydrogen bonding, namely PEGylated Au nanoparticles [50]. PEGylated Au nanoparticles own the capacity for anti-immune response as well as an extended retention time in the bloodstream while inhibiting the reticuloendothelial system (RES) to uptake [51]. Au nanoparticles conjugated with peptides or antibodies were able to demonstrate the regulation of cell cycle progression to induce cancer cell apoptosis while preventing the healthy surrounding tissue from experiencing any side effects [52]. Furthermore, EDC (1-ethyl-3-dimethylaminopropyl carbodiimide hydrochloride) [53] was used as a cross-linking agent to combine BP with PEG-Au nanocarriers in the current study. EDC reacted with the carboxyl group (-COOH) to form a hyperactive intermediate. Next, the intermediate reacted with the primary amine to form Amide bonding [54], which corresponded to the FTIR result (Figure 1C). In the present study, Au nanoparticles were used as an inorganic core to combine with PEG, which, when applied as stable carriers to cross-link with BP, successfully further investigated the delivery efficiency within DBTRG human glioma cells.

n-butylidenephthalide (BP) is a natural bioactive compound with superior anti-cancer capacities for fighting various cancer diseases [55,56]. BP has been verified to inhibit tumor cell proliferation by regulating both cell cycle progression and endocytosis efficiency to further achieve cell apoptosis [57]. In brief, the available literature has classified the major types of cell death progression: apoptosis, necrosis, oncosis, pyroptosis, and autophagy [58]. Cell apoptosis is a programmed cell death induced through a series of molecular pathways due to the occurrence of physiological or pathological factors [59]. Progression of apoptosis is associated with two types of caspases: initiator caspases and executioner caspases [60]. The initiator caspases (caspases 8 and 9) are activated due to the cell being destroyed. Afterwards, initiator caspases further activate the executioner caspases (caspases 3, 6, and 7). The executioner caspases then activate the endonucleases to process DNA fragmentation [61]. The contents of the apoptotic cells are phagocytosed by neighboring cells. Cell necrosis is uncontrolled cell death progression suddenly stimulated by severe inflammation [60]. Once the cell has been seriously destroyed by external attacks, such as hypoxia, the necrosis pathway is activated by proinflammatory proteins (NF-κB). The cell body will begin swelling until the cell membrane bursts, with the released organelles stimulating an inflammatory response that is then phagocytosed by white blood cells. According to previous research, apoptotic and necrotic cells can be detected by Annexin-V/PI double staining with flow cytometry. In the present study, DBTRG human glioma cells were treated with various nanoparticles, then subjected to Annexin-V/PI double staining with flow cytometry detection (Figure 7C,D). Annexin-V-positive cells are represented as apoptotic DBTRG cells, where the quantified result is 31.5% in the PEG-Au-BP group, which is more than the 19.3% in the BP group. The average population of dead (necrotic) cells was 13.3% in the PEG-Au-BP group, which is less than the 17.2% in the BP group. The above evidence indicates that PEG-Au-BP exhibits a better induction of cell apoptosis when compared to pure BP. Furthermore, the fewer number of dead (necrotic) cells after PEG-Au-BP treatment may suggest a reduction in inflammatory response, which corresponds to the results published in the previous literature.

Consequently, BP could arrest the cell cycle for proliferation and activate glioblastoma cell apoptosis through inducing the expression of the orphan nuclear receptor. Subsequently, the cytochrome c could be released and undergo apoptosis relating to executioner caspase-3 protein [62]. Previous studies have prepared polycationic liposomal polyethylenimine and polyethylene glycol (PEG) complex (LPPC) to encapsulate BP. DBTRG cells were applied to be treated with the BP-LPPC drug, with the results demonstrating that cell uptake efficiency became higher. In addition, BP-LPPC could arrest the cell cycle at the Sub-G1 phase to activate cell apoptosis [63]. According to the results from the present research, the cell population in the Sub-G1 phase was 37.25% in the PEG-Au-BP group, and 37.25% in the BP group. This evidence indicates that PEG-Au carrying BP has the better ability to induce DBTRG cell apoptosis when compared to BP alone (Figure 7A,B).

The apoptotic pathways induced by a natural BP compound has also been investigated in previous studies. DBTRG human glioblastoma multiforme cells were treated with BP, where the expression of p21, Bax, and active caspase-3 protein was found to be induced in the DBTRG cells after BP treatment [64]. Other studies have treated DBTRG cells with BP-LPPC, with those results demonstrating that BP-LPPC facilitated p21, Bax, and caspase-3 expression, while reducing Cyclin D expression [63]. The expression of p21 in cells leads to G1 phase arrest by binding to Cyclin-D1-CDK complex [65]. Furthermore, BP-LPPC induced overexpression of p53, which activates Bax and further stimulates the release of cytochrome c to active caspase-3 protein, which is also associated with cell apoptosis [63]. In addition, a previous study provided evidence that BP could induce Sub-G1 arrest in human bladder cancer cells [66]. This suggests that PEG-Au-BP is a potential nanodrug for DBTRG cell apoptosis, which corresponds to our results seen in Figure 8.

Cell uptake plays an important role in the different particle sizes of drug entry into cells. Endocytic routes can be categorized into several types: micropinocytosis, clathrin-mediated endocytosis, cell autophagy, and phagocytosis [67]. Therefore, the mechanisms of endocytic routes were investigated by the various inhibiters. Based on a previous study, the uptake of BP-LPPC in DBTRG cells was inhibited by Chlorpromazine (CPZ) [63], which could inhibit clathrin-mediated endocytosis by anchoring clathrin and Adaptor Protein 2 (AP2) complex to endosomes to form coated pits for delivery [68]. In the present research, the uptake of PEG-Au-BP nanoparticles in DBTRG cells was inhibited by CPZ and Bafilomycin (Baf). Baf has been verified to downregulate vacuolar ATPase inhibitors, which is related to cell autophagy [69]. Thus, the favorable routes where DBTRG cells “eat” PEG-Au-BP nanoparticles are related to clathrin-mediated endocytosis and cell autophagy (Figure 6). Furthermore, lysosomes are the organelle needed to undergo endocytosis. As mentioned above, cell autophagy is one of the mechanisms for DBTRG cells to uptake PEG-Au-BP. Lysosome and autophagosome are associated with autophagy in processing protein degradation in an acidic environment (pH 4.5–5.0) [70,71]. However, according to the LysoTracker assay in the present research, the PEG-Au-BP nanoparticles in DBTRG cells were observed to be stable after internalization in lysosomes (Figure 5). This suggests that PEG-Au-BP is a suitable candidate for brain cancer treatment.

PEG-Au-BP nanodrugs were also prepared for the in vivo assessment. Indeed, BP has been proven to penetrate the BBB to reach the target site for purposes of inhibiting malignant brain tumor growth in mice [64]. Based on our animal model, PEG-Au-BP nanoparticles could penetrate the BBB, while demonstrating an extended retention period at 12 and 24 h (Figure 9C and Appendix A). In closing, PEG-Au-BP nanoparticles are a potential nanodrug delivery system owing to their better cytotoxicity, biocompatibility, stability, and enhancement of uptake efficiency, all of which makes for a promising nanomedicine for targeting human brain cancer in clinical treatments.

## 5. Conclusions

The present research prepared PEG-Au nanocarriers cross-linked with BP to investigate both its efficacy in anti-brain-cancer capacity and retention period in brain tissue. According to the results taken from the biocompatibility and biological function assessments, PEG-Au-BP nanoparticles could significantly reduce any proliferation, as well as induce cell apoptosis in DBTRG brain cancer cells. Furthermore, mice models elucidated that PEG-Au-BP would not cause damage to tissue integrity in the brain (forebrain, cerebellum, and midbrain), heart, liver, spleen, lung, and kidney. Meanwhile, biodistribution assessments indicated that PEG-Au-BP could penetrate the blood-brain barrier (BBB) and remain retained in the brain for an extended retention period. The above findings suggest that PEG-Au-BP nanodrugs could well be an innovative drug delivery system for clinical anti-brain-cancer therapeutics.

## Figures and Tables

**Figure 1 cells-11-02172-f001:**
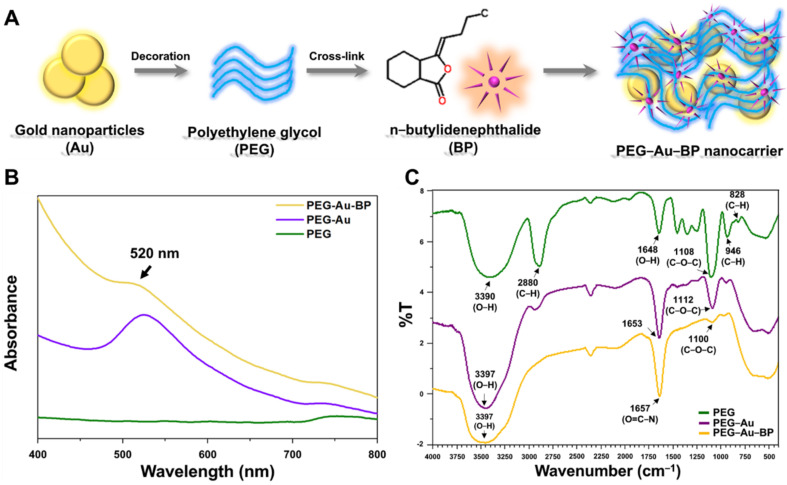
Characterization of PEG-Au nanocarriers cross-linked with BP. (**A**) Concept illustration for the preparation of polyethylene-glycol-gold nanoparticles with n-butylidenephthalide (PEG-Au-BP) nanodrugs. PEG was first combined with Au nanoparticles through sonication. Next, the as-prepared PEG-Au nanocarriers were cross-linked with BP to obtain PEG-Au-BP. (**B**) The UV-Vis spectra demonstrated the presence of Au nanoparticles in PEG-Au and PEG-Au-BP with an absorption peak at 520 nm. (**C**) FTIR spectra indicate the specific functional groups in pure PEG, PEG-Au, and PEG-Au-BP at the total wavenumber from 400 cm^−1^ to 4000 cm^−1^.

**Figure 2 cells-11-02172-f002:**
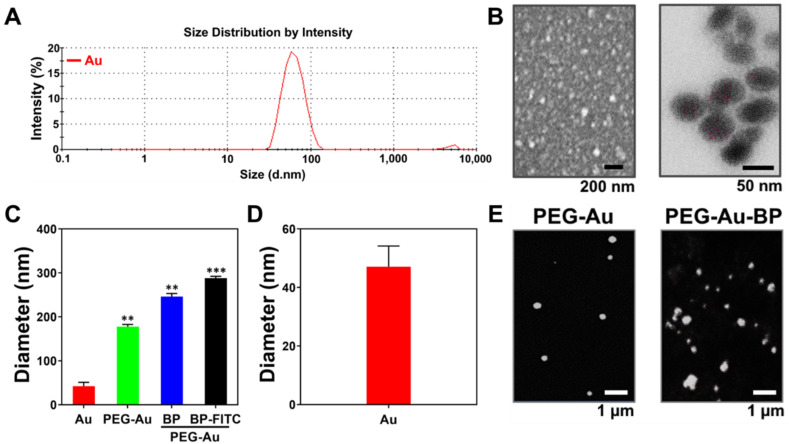
Identification for the size of various nanoparticles. (**A**) The size distribution intensity of Au nanoparticles was measured by DLS assay. (**B**) The left panel demonstrates the SEM image of Au nanoparticles. The scale bar equals 200 nm. The right panel displays the TEM image of Au nanoparticles. The scale bar = 50 nm. (**C**) The diameter of various nanoparticles detected by DLS assay. The size of Au, PEG-Au, PEG-Au-BP, and FITC-labeled PEG-Au-BP was demonstrated as 42.02 ± 8.95 nm, 177.5 ± 5.2, 246.27 ± 7.0, and 288.2 ± 4.2, respectively. ** *p* < 0.01, *** *p* < 0.001: compared to Au nanoparticle. (**D**) The diameter of Au nanoparticle was quantified and the average size of Au nanoparticles was 44 nm. (**E**) SEM images for the PEG-Au and PEG-Au-BP nanoparticles. The scale bar = 1 μm. Results are presented as one of three independent experiments.

**Figure 3 cells-11-02172-f003:**
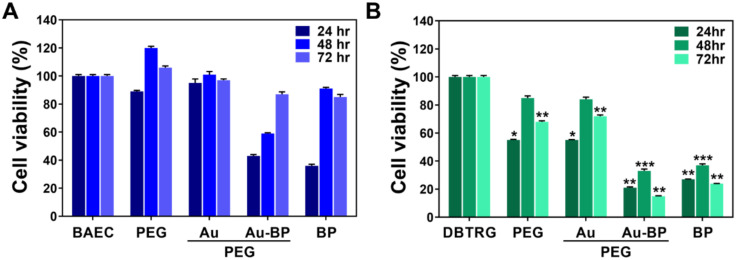
Biocompatibility assessment for BAEC and DBTRG cell lines investigated by MTT assay. Cytotoxicity test between (**A**) BAEC and (**B**) DBTRG brain cancer cell lines treated with pure PEG, PEG-Au, PEG-Au-BP, and pure BP for 24, 48, and 72 h. The semi-quantified result of BAEC cell growth was inhibited by PEG-Au-BP and pure BP, but with no significant difference. However, the cell viability of DBTRG brain cancer cells was significantly lower in PEG-Au-BP and pure BP treatments compared to control, indicating that the PEG-Au-BP could be a potential nanodrug to inhibit cancer cell proliferation. * *p* < 0.05, ** *p* < 0.01, *** *p* < 0.001: compared to the control group (BAEC and DBTRG).

**Figure 4 cells-11-02172-f004:**
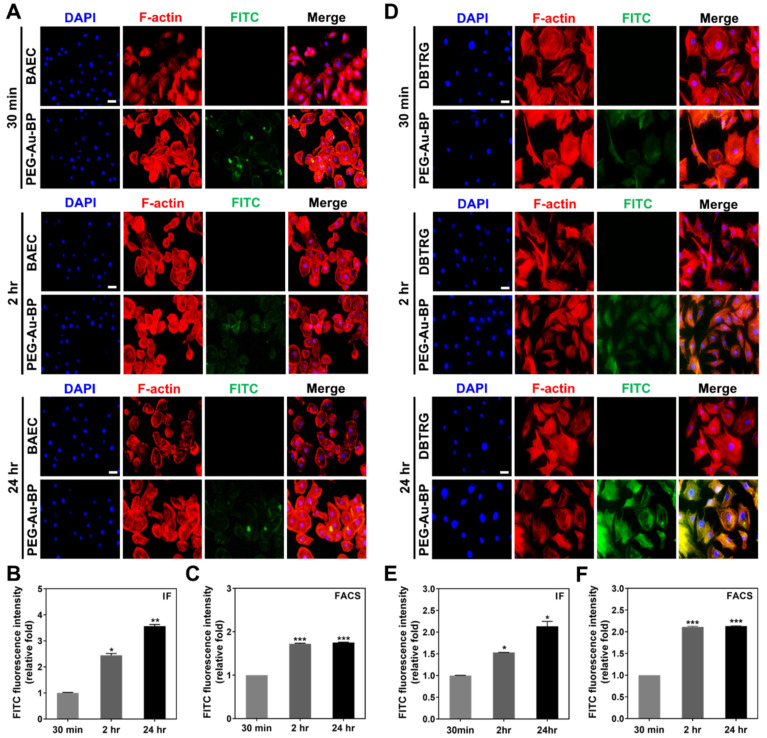
Cell uptake ability in BAEC and DBTRG cell lines treated with PEG-Au-BP. PEG-Au-BP nanoparticles were labeled with fluorescein (FITC) to observe the cell transportation through a fluorescent microscope. The immunofluorescent (IF) images were demonstrated as (**A**) BAEC and (**D**) DBTRG cell lines, including at 30 min, 2 h, and 24 h. Based on the FITC images (green color), the uptake amount at 2 h in both cell lines was discovered to be higher than 30 min, with the 24 h group even higher than the other two groups. Furthermore, the uptake efficiency of both cell lines was semi-quantified by the immunofluorescence (IF) and fluorescence-activated cell sorting (FACS) methods. (**B**,**C**) The results from both the IF and FACS methods indicate that the uptake amount of PEG-Au-BP in BAEC significantly increased at 2 h and 24 h. (**E**,**F**) In DBTRG brain cancer cell lines, the semi-quantification results show the remarkably increased amount at 2 h and 24 h in both the IF and FACS methods compared to the 30 min group. Results are demonstrated as one of three independent experiments. DAPI was applied to locate cell nuclei (blue color). Red color fluorescent presented as F-actin. The scale bar equals 20 μm. * *p* < 0.05, ** *p* < 0.01, *** *p* < 0.001: compared to the 30 min group.

**Figure 5 cells-11-02172-f005:**
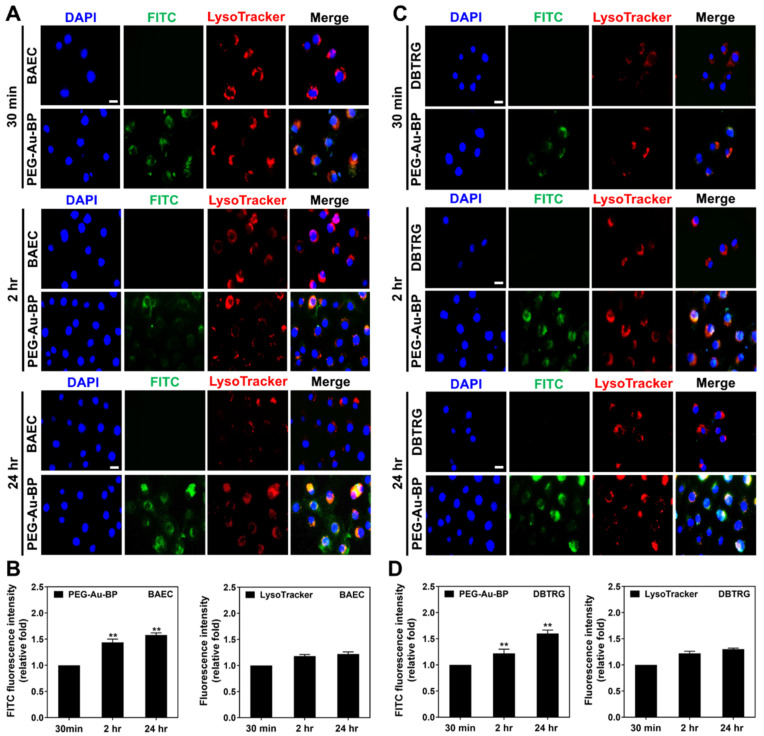
Inside cell transportation for PEG-Au-BP nanodrugs in BAEC and DBTRG cell lines. Lysosome is the major organelle for metabolizing foreign bodies to entry cells. Therefore, LysoTracker assay was used to detect the lysosomes (red color) for potential transportation combined with the green color fluorescence of FITC-labeled PEG-Au-BP nanoparticles, while the images of (**A**) BAEC and (**C**) DBTRG cells demonstrated the lysosomes mainly located at the perinuclear site. (**B**) Based on the quantified results from the IF method, the FITC-labeled PEG-Au-BP intensity in BAEC significantly increased at 2 h (~1.44-fold) and 24 h (~1.58-fold). (**D**) The results of DBTRG brain cancer cells also show that the FITC-labeled PEG-Au-BP uptake amount increased to ~1.22-fold at 2 h and ~1.60-fold at 24 h compared to the 30 min group. DAPI solution was used to stain cell nuclei (blue color). Scale bar equals to 10 μm. ** *p* < 0.01: compared to the 30 min group.

**Figure 6 cells-11-02172-f006:**
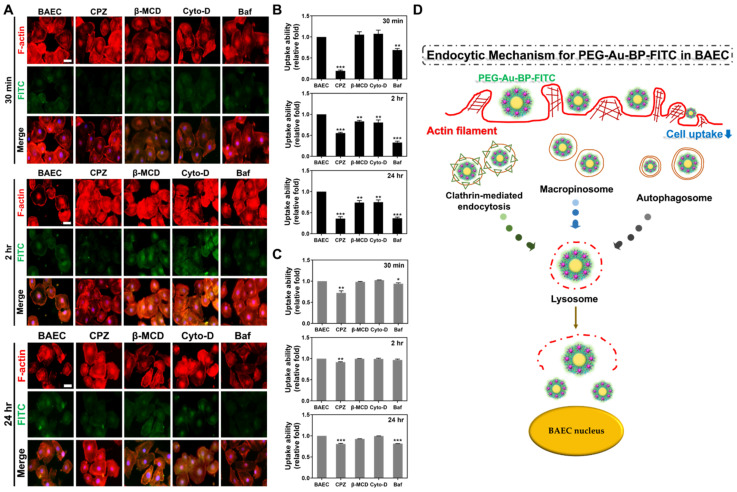
Investigation of endocytic routes in BAEC and DBTRG cell lines. The endocytosis mechanisms were caveolae, macropinocytosis, receptor-mediated endocytosis, and phagocytosis. Afterwards, four different endocytic inhibitors, CPZ, β-MCD, Cyto-D, and Baf, were treated with both cell lines at different time points (30 min, 2 h, and 24 h) to investigate the cell entry process. (**A**) The immunofluorescence images of BAEC uptake of PEG-Au-BP at each time point. (**B**) The quantitative results of fluorescence intensity based on the IF method indicate that PEG-Au-BP colocalized with F-actin was significantly reduced after being treated with CPZ (30 min (~0.19-fold), 2 h (~0.55-fold), 24 h (~0.36-fold)) and Baf (30 min (~0.68-fold), 2 h (~0.32-fold), 24 h (~0.36-fold)). (**C**) Furthermore, the fluorescein-positive cells semi-quantified by the FACS method significantly decreased when influenced by CPZ (30 min (~0.72-fold), 2 h (~0.91-fold), 24 h (~0.80-fold)) and Baf inhibitors (30 min (~0.94-fold), 2 h (~0.96-fold), 24 h (~0.81-fold)) at each time point. (**D**) Schematic illustration of endocytic routes for normal BAEC. (**E**) The immunofluorescence images of DBTRG brain cancer cells uptake of PEG-Au-BP at each time point. (**F**) According to the results quantified by the IF method, the uptake amount in DBTRG significantly decreased in CPZ (30 min (~0.29-fold), 2 h (~0.28-fold), 24 h (~0.34-fold)) and Baf (30 min (~0.21-fold), 2 h (~0.29-fold), 24 h (~0.34-fold)) treatments. (**G**) The uptake amount quantified by the FACS method demonstrated a similar trend with the IF method, while the CPZ (30 min (~0.71-fold), 2 h (~0.87-fold), 24 h (~0.88-fold)) and Baf (30 min (~0.91-fold), 2 h (~0.97-fold), 24 h (~0.89-fold)) group was remarkably lower than the control group. (**H**) Schematic illustration of endocytic routes for DBTRG brain cancer cells. Results were represented as the mean ± SD (*n* = 3). Red color: F-actin. Blue color: cell nuclei. Green color: FITC-labeled PEG-Au-BP. The scale bar equals 20 μm.* *p* < 0.05, ** *p* < 0.01, *** *p* < 0.001: compared to the 30 min group.

**Figure 7 cells-11-02172-f007:**
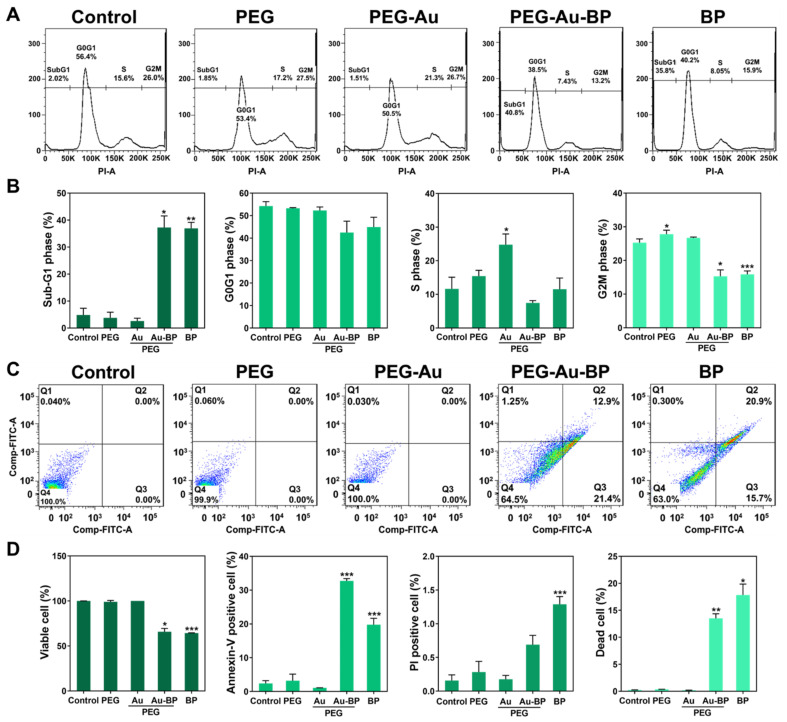
Cell apoptosis of DBTRG brain cancer cells after being treated with various nanoparticles for 48 h. (**A**) The cell cycle histograms of DBTRG cells affected by PEG-Au-BP were obtained from flow cytometry. (**B**) Sub-G1 phase indicates the cell apoptosis. Based on the quantification results, PEG-Au-BP (37.25%) and BP (36.92%) treatments had the higher percentage of apoptotic DBTRG cells than the other groups. To further determine the population of apoptotic DBTRG brain cancer cells after 48 h of each treatment, Annexin-V/PI double staining assay was applied for the investigation. (**C**) The histograms obtained by flow cytometry were demonstrated. (**D**) The population was further quantified, while the viable cells in the PEG-Au-BP group (63.2%) and BP group (61.7%) were significantly lower than the other groups. The apoptotic cells as well as Annexin-V-positive cells in the PEG-Au-BP group (31.5%) and BP group (19.3%) were remarkably higher than the others. The above evidence indicates that PEG-Au-BP could be a potential anti-cancer nanodrug. The data were represented as the mean ± SD (*n* = 3). * *p* < 0.05, ** *p* < 0.01, *** *p* < 0.001: compared to the control group.

**Figure 8 cells-11-02172-f008:**
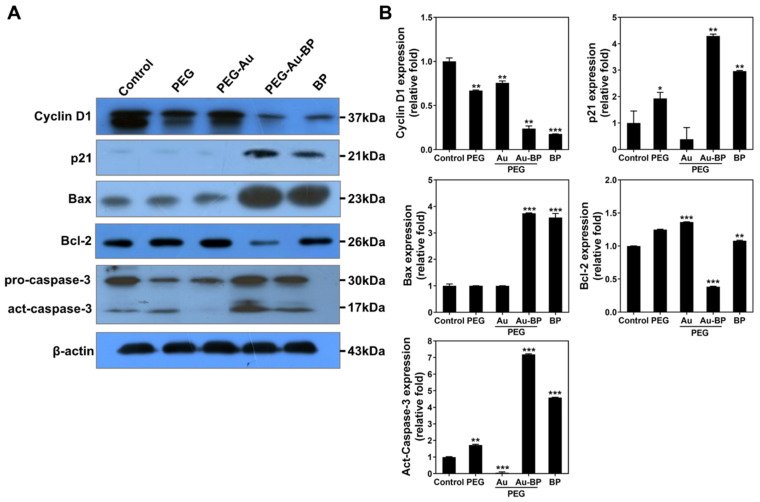
Expression of apoptotic-related proteins in DBTRG brain cancer cells after 48 h of various treatments. (**A**) The Western blotting images of various proteins in each treatment were displayed. (**B**) The protein expression was semi-quantified. The expression of apoptotic cascade proteins p21, Bax and active caspase-3 was significantly higher in the PEG-Au-BP group (p21: ~4.29 fold, Bax: ~3.73 fold, Act-caspase-3: ~7.19-fold), indicating PEG-Au-BP could strengthen the apoptosis effect in the DBTRG cells line. Results were presented as the mean ± SD (*n* = 3). * *p* < 0.05, ** *p* < 0.01, *** *p* < 0.001: compared to the control group.

**Figure 9 cells-11-02172-f009:**
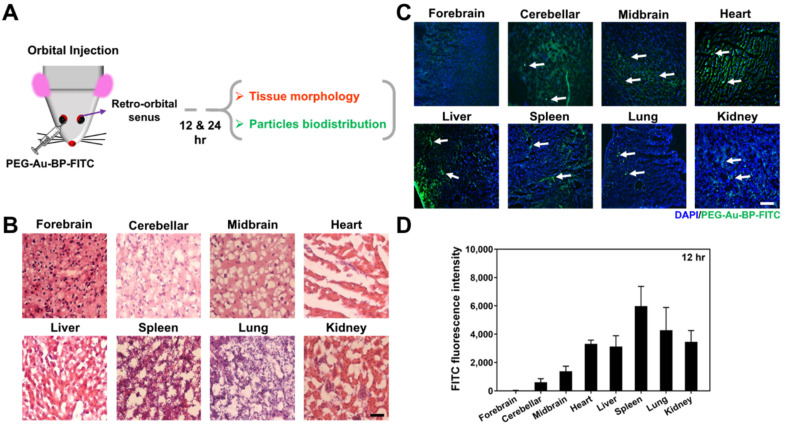
Tissue morphology and particle distribution after PEG-Au-BP treatment for 12 h in an animal model. (**A**) The illustration indicated the mice were injected with PEG-Au-BP nanoparticles through retro-orbital sinus injection. (**B**) Each tissue was stained with H&E staining to observe the tissue integrity after the treatment. The images indicated that the organs would not be harmed by PEG-Au-BP. (**C**) The histological images demonstrated the particle distribution in each organ. The white arrows demonstrate the FITC labeled PEG-Au-BP nanoparticles (green color). Cell nuclei were stained by DAPI solution (blue color). Scale bar equals to 50 um. (**D**) The FITC fluorescence intensity of PEG-Au-BP in each organ was quantified, where the nanoparticles could also be observed in brain tissue, indicating PEG-Au-BP could stay in brain tissue for anti-cancer treatment.

**Figure 10 cells-11-02172-f010:**
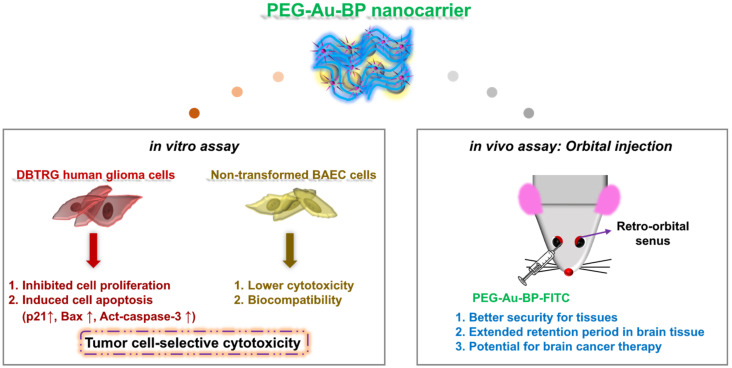
Schematic illustration for PEG-Au-BP nanocarrier through in vivo and in vitro assessments. The as-prepared PEG-Au-BP nanoparticles were applied to treat DBTRG human glioma cells and non-transformed BAEC cells. Based on the in vitro results, PEG-Au-BP could remarkably inhibit the proliferation of DBTRG brain cancer cells, as well as induce cell apoptosis. Furthermore, the evidence from in vivo assay strongly suggested PEG-Au-BP to be a bio-safety nanodrug for brain cancer treatments.

## Data Availability

Data are contained within the article.

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
