# Peer review of "Improved Delivery Performance of n-Butylidenephthalide-Polyethylene Glycol-Gold Nanoparticles Efficient for Enhanced Anti-Cancer Activity in Brain Tumor"

_cells, 2022, doi:10.3390/cells11142172_

Round 1
Reviewer 1 Report
Comments:
1. TEM images of PEG-Au and PEG-Au-BP should be provided.
2. In Figure 3, the cell viability is used as the label of the ordinate, but the actual value is the OD value at 570nm, which is easy to be misunderstood. In addition, this data display method cannot directly reflect the cytotoxicity of drugs. Please revise it.
3. The author mentioned that BP is a natural bioactive compound with superior anti-cancer capacities. So the PEG-Au-BP belongs to a new nanoformulation of BP, why the author believes it is a nanocarrier (Line 391)?
4. What is the function of Au? The author did not confirm its role in the PEG-Au-BP?
5. What is the mechanism of PEG coating on Au-NPs? Although the reference 48 believed that Au nanoparticles could be modified by PEG with the thiol bonding. However, in your work, we did not find the thiol bonding in PEG or PEG-Au? The author should explain it in the discussion.
6. In your developed PEG-Au-BP, what is the loading of Au and BP?
Author Response
Reviewer 1
Comments:
- TEM images of PEG-Au and PEG-Au-BP should be provided.
Answer:
Thanks for the comment from the reviewer.
(1) We apologize for the inconvenience to provide the TEM images. The TEM instrument was not available for PEG-Au and PEG-Au-BP observation due to COVID-19 epidemic in our country now. However, we have provided the new SEM data for the nanoparticle observation of PEG-Au and PEG-Au-BP and further confirmed the nanosize. Based on this finding, we will continuous execute the in vivo animal experiment to investigate the anti-brain cancer capacity of PEG-Au-BP nanoparticles. Therefore, we hope reviewer understand the TEM data of PEG-Au and PEG-Au-BP will be included in the new research paper.
(2) The SEM images were included in new Figure 2, and the new description was described in the section 3.1.. “Both size distribution intensity and size of various nanoparticles were investigated through a DLS analyzer. The percentage of intensity is demonstrated as Figure 2A & S1. The Au nanoparticles were also observed by SEM (Figure 2B-Lt. panel) and TEM (Figure 2B-Rt. panel). Additionally, the diameter of Au, PEG-Au, PEG-Au-BP and FITC labeled PEG-Au-BP measured by DLS assay was 42.02±8.95 nm, 177.5±5.2 nm, 246.2±7.0 nm and 288.2±4.2 nm, respectively (Figure 2C). The average size of Au nanoparticles was quantified as 44 nm through the SEM (Figure 2D), which corresponded to the results of DLS analysis. Moreover, the SEM images of PEG-Au and PEG-Au-BP nanoparticles were demonstrated as Figure 2E. Each nanoparticle was subjected to further experiments.” (Page 7, Line 329-337) - In Figure 3, the cell viability is used as the label of the ordinate, but the actual value is the OD value at 570 nm, which is easy to be misunderstood. In addition, this data display method cannot directly reflect the cytotoxicity of drugs. Please revise it.
Answer:
Thanks for the valuable suggestion from the reviewer. We have revised the presentation of cell viability into percentage in the Figure 3 and the description in the section 3.2. “The cytotoxicity of various nanomaterials was investigated through MTT assay after 24, 48 and 72 hours of incubation. In the non-transformed BAEC cell line (Figure 3A), the cell viability (%) for 24-hour incubation is demonstrated [Control group (BAEC): 100%, PEG group: 89%, PEG-Au group: 95%, PEG-Au-BP group: 43%, and BP group: 36%]. For 48-hour incubation, the results are presented as Control group (BAEC): 100%, PEG group: 120%, PEG-Au group: 101%, PEG-Au-BP group: 59%, and BP group: 91%. Regarding 72-hour incubation, the data revealed Control group (BAEC): 100%, PEG group: 106%, PEG-Au group: 97%, PEG-Au-BP group: 87%, and BP group: 85% (Figure 3A). Furthermore, the cytotoxicity of brain cancer DBTRG cell line, as induced by various materials was examined. The cell viability (%) results at 24, 48 and 72 hours are displayed as follows: 24 hours [Control group (DBTRG): 100%, PEG group: 55%, PEG-Au group: 55%, PEG-Au-BP group: 21%, and BP group: 27%]; 48 hours: [Control group (DBTRG): 100%, PEG group: 85%, PEG-Au group: 84%, PEG-Au-BP group: 33%, and BP group: 37%]; 72 hours: [Control group (DBTRG): 100%, PEG group: 68%, PEG-Au group: 72%, PEG-Au-BP group: 15%, and BP group: 24%] (Figure 3B). The above results demonstrate that PEG-Au-BP could significantly inhibit DBTRG brain cancer cell proliferation when compared to the PEG and PEG-Au groups.” (Page 9, Line 363-379) - The author mentioned that BP is a natural bioactive compound with superior anti-cancer capacities. So the PEG-Au-BP belongs to a new nanoformulation of BP, why the author believes it is a nanocarrier (Line 391)?
Answer:
Thanks for the comment from the reviewer. We have revised the description in the caption of Figure 3. “However, the cell viability of DBTRG brain cancer cells was significantly lower in PEG-Au-BP and pure BP treatments compared to control group, indicating that the PEG-Au-BP could be a potential nanodrug to inhibit cancer cell proliferation.” (Page 9, Line 382-385) - What is the function of Au? The author did not confirm its role in the PEG-Au-BP?
Answer:
Thanks for the valuable comment from the reviewer.
We have included the description of the Au function in the “Discussion” section.
(1) “Gold nanoparticles (Au) are the most promising candidate to be the inorganic core for the drug delivery system, owing to their superior biocompatibility and ease in functionalizing with bio-active molecules [46,47].” (Page 18, Line 629-631)
(2) “In the present study, Au nanoparticles were used as inorganic core to combine with PEG, which when applied as stable carriers to cross-link with BP, successfully further investigated the delivery efficiency within DBTRG human glioma cells.” (Page 19, Line 644-647)
Reference:
46. Arvizo, R.; Bhattacharya, R.; Mukherjee, P. Gold nanoparticles: opportunities and challenges in nanomedicine. Expert opinion on drug delivery 2010, 7, 753-763
47. Li, W.; Cao, Z.; Liu, R.; Liu, L.; Li, H.; Li, X.; Chen, Y.; Lu, C.; Liu, Y. AuNPs as an important inorganic nanoparticle applied in drug carrier systems. Artificial cells, nanomedicine, and biotechnology 2019, 47, 4222-4233. - What is the mechanism of PEG coating on Au-NPs? Although the reference 48 believed that Au nanoparticles could be modified by PEG with the thiol bonding. However, in your work, we did not find the thiol bonding in PEG or PEG-Au? The author should explain it in the discussion.
Answer:
Thanks for the valuable comment from the reviewer. We have deleted the description related to the thiol bonding in the “Discussion” section, and the new description was included. Additionally, we have also revised the FTIR spectrum in the Figure 1C.
(1) “The specific peaks of PEG are seen at 3390 cm-1 (-OH stretching), 2880 cm-1 (C-H stretching), 1648 cm-1 (-OH blending), and 1108 cm-1 (C-O-C stretching) [38]. After PEG conjugated with Au, the peak of 3390 cm-1 (-OH bond) shifted to 3397 cm-1, indicating Au nanoparticles were combined with PEG (Figure 1C). Furthermore, the peaks of 1653 cm-1 shifted to 1657 cm-1 (O=C-N, Amide bond), showing PEG-Au was cross-linked with BP (Figure 1C).” (Page 7, Line 323-328)
Reference:
38. Loloei, M.; Omidkhah, M.; Moghadassi, A.; Amooghin, A.E. Preparation and characterization of Matrimid® 5218 based binary and ternary mixed matrix membranes for CO2 separation. International Journal of Greenhouse Gas Control 2015, 39, 225-235.
(2) “According to previous literature, the stability of Au nanoparticles could be modified by PEG with the hydrogen bonding, also namely PEGylated Au nanoparticles [50]. PEGylated Au own the ability of anti-immune response as well as have an extended retention time in the bloodstream, while inhibiting the reticuloendothelial system (RES) to uptake [51].” (Page 18-19, Line 633-637)
Reference:
50. Stiufiuc, R.; Iacovita, C.; Nicoara, R.; Stiufiuc, G.; Florea, A.; Achim, M.; Lucaciu, C.M. One-step synthesis of PEGylated gold nanoparticles with tunable surface charge. Journal of Nanomaterials 2013, 2013.
51. Perrault, S.D.; Walkey, C.; Jennings, T.; Fischer, H.C.; Chan, W.C. Mediating tumor targeting efficiency of nanoparticles through design. Nano letters 2009, 9, 1909-1915.
(3) “Furthermore, EDC (1-ethyl-3- dimethylaminopropyl carbodiimide hydrochloride) [53] was used as cross-linking agent to combine BP with PEG-Au nanocarriers. EDC would react with carboxyl group (-COOH) to form hyperactive intermediate. Next, the intermediate would react with primary amine to form amide bonding [54], which corresponded to the FTIR result (Figure 1C).” (Page 19, Line 640-644)
Reference:
53. Grabarek, Z.; Gergely, J. Zero-length crosslinking procedure with the use of active esters. Analytical biochemistry 1990, 185, 131-135.
54. Nakajima, N.; Ikada, Y. Mechanism of amide formation by carbodiimide for bioconjugation in aqueous media. Bioconjugate chemistry 1995, 6, 123-130. - In your developed PEG-Au-BP, what is the loading of Au and BP?
Answer:
Thanks for the comment from the reviewer. We have revised the description in the Section 2.1. (Page 3-4, Line 143-156) “PEG solution was purchased from Sigma-Aldrich, USA (average mol wt = 200 kDa). 1 ml of PEG solution (500 µM) was diluted with 24 ml of deionized water to obtain 20 µM PEG solution. Afterwards, 609.6 µl of PEG solution was mixed with 390.4 µl of 50 ppm physical gold nanoparticle solution (Gold NanoTech, Inc, Taiwan) for 2 hours to obtain the PEG-Au solution. n-Butylidenephthalide (BP) solution was purchased from Alfa Aesar, Ward Hill, NY, USA (10 mg/ml, 95%, mol weight: 188.23). To prepare the PEG-Au-BP solution, BP solution was firstly mixed with 1-ethyl-3-(3-dimethyl aminopropyl)-carbodiimide solution (EDC, 15 mM, ThermoFisher, USA) at volume ratio 1:2 to interact for 2 hours at room temperature. 6 µl of mixture was then combined with 250 µl of PEG-Au solution to react at 4oC for 8 hours. The PEG-Au-BP solution with 50 ppm of Au nanoparticles and 78 µg/ml of BP was obtained. Furthermore, to investigate the cellular metabolic process, 0.5 mg/ml of fluorescein isothiocyanate reagent (FITC-conjugated AffiniPure Goat Anti-Rabbit IgG, Jackson ImmunoResearch, USA) was conjugated with PEG-Au-BP through coupling with the amine groups at 4oC for 8 hours in dark at a 50:1 volume ratio.”

Reviewer 2 Report
n-butylidenephthalide (BP) has been verified for cancer cell toxicity. In this project, Au nanoparticles were firstly conjugated with Polyethylene Glycol (PEG). Then, they cross-linked them with BP to obtain PEG-Au-BP nanocarriers. The physicochemical properties were characterized through Ultraviolet-visible spectroscopy (UV-Vis), Fourier-transform Infrared spectroscopy (FTIR), and Dynamic Light Scattering (DLS) were used to confirm the combination of PEG, Au, and BP. According to the results of the MTT assay, PEG-Au-BP was able to significantly inhibit brain cancer cells. The authors demonstrate efficacy against cancer cells. Tissue integrity in the brain (forebrain, cerebellar, and midbrain), heart, liver, spleen, lung, and kidney, did not show significant destruction due to PEG-Au-BP treatment.
Flow cytometry scatters should be improved. Thank you for the WB genuine bands.
We need to see more tissue integrity in all organs and in many fields to allow that there is no significant destruction. Supplementary data can be provided.
English needs to be improved.
Do the results conclude ??? The authors conclude I guess, they should write in the paper.
Author Response
Reviewer 2.
n-butylidenephthalide (BP) has been verified for cancer cell toxicity. In this project, Au nanoparticles were firstly conjugated with Polyethylene Glycol (PEG). Then, they cross-linked them with BP to obtain PEG-Au-BP nanocarriers. The physicochemical properties were characterized through Ultraviolet-visible spectroscopy (UV-Vis), Fourier-transform Infrared spectroscopy (FTIR), and Dynamic Light Scattering (DLS) were used to confirm the combination of PEG, Au, and BP. According to the results of the MTT assay, PEG-Au-BP was able to significantly inhibit brain cancer cells. The authors demonstrate efficacy against cancer cells. Tissue integrity in the brain (forebrain, cerebellar, and midbrain), heart, liver, spleen, lung, and kidney, did not show significant destruction due to PEG-Au-BP treatment.
- Flow cytometry scatters should be improved. Thank you for the WB genuine bands.
Answer:
Thanks for the valuable comment from the reviewer. We have provided the better quality of flow cytometry scatters in the Figure 7A & 7C. (Page 15) - We need to see more tissue integrity in all organs and in many fields to allow that there is no significant destruction. Supplementary data can be provided.
Answer:
Thanks for the valuable suggestion from the reviewer.
(1) We have included the more data in the Supplementary materials (Figure S4), demonstrating as the PEG-Au nanoparticles in stomach and spinal cord tissue after 12 and 24 hours treatments and the tissue integrity seems well and no significant destruction.
(2) The description was included in section 3.7. (Page 17, Line 581-583)
“Additionally, we also provide that PEG-Au-BP nanodrug could be observed in stomach and spinal cord tissue (Figure S4).” - English needs to be improved.
Answer:
Thanks for the suggestion from the Reviewer. We have revised some sentences in the present research.
(1) “…non-transformed BAEC and DBTRG human glioma cells were treated with PEG-Au-BP drugs to investigate the tumor-cell selective cytotoxicity…” (Page 1, Line 35-36)
(2) “…determined to be favorable routes for BAEC and DBTRG cells to absorb PEG-Au-BP nanodrugs.” (Page 1, Line 41-42)
(3) “Through in vivo assessments, the tissue morphology and particle distribution in a mice model were examined after a retroorbital sinus injection containing PEG-Au-BP nanodrugs. ” (Page 1-2, Line 48-50)
(4) “Simultaneously, the extended retention period for PEG-Au-BP nanodrugs was discovered, particularly in brain tissue.” (Page 2, Line 52-53)
(5) “n-butylidenephthalide (denoted as BP in the current research) is a single purified compound extracted from Angelica using chloroform and having a molecular weight of 188.23, with the molecular formula of C12H12O2 [22].” (Page 3, Line 99-102)
(6) “BP has been also demonstrated as owning the powerful ability for cancer cell toxicity [25], and inducing stem cell differentiation at a low dose [26].” (Page 3, Line 104-106)
(7) “In this research, biocompatible PEG-Au nanocarriers cross-linked with BP were prepared.” (Page 3, Line 136-137)
(8) “Figure 1A indicates the brief procedure performed for preparation of polyethylene glycol-gold nanoparticles cross-linked with n-butylidenephthalide (PEG-Au-BP). Afterwards, the physicochemical properties of the PEG-Au-BP nanoparticles have been characterized through various methods.” (Page 3, Line 318-321)
(9) “Both BAEC and DBTRG cell lines were subjected to an investigation of cell uptake efficiency after being treated with FITC labeled PEG-Au-BP nanodrugs.” (Page 9, Line 389-390)
(10) “Furthermore, PEG-Au-BP nanoparticles uptake by DBTRG brain cancer cells was also examined based on the IF and FACS methods.” (Page 10, Line 399-401)
(11) “The results indicate that PEG-Au-BP nanoparticles could be significantly uptaken by BAEC and DBTRG brain cancer cells.” (Page 10, Line 405-407)
(12) “Therefore, the above LysoTracker results elucidate that after autophagy, PEG-Au-BP nanodrugs would not degrade in lysosome, which remained stable in both cell lines. This indicates that PEG-Au-BP could be potential nanoparticles for drug delivery systems.” (Page 11, Line 433-436)
(13) “…the endocytotic routes in both the BAEC and DBTRG brain cancer cell lines for absorbing PEG-Au-BP nanoparticles…” (Page 11, Line 475-477)
(14) “The illustration indicated the mice were injected with PEG-Au-BP nanoparticles through retroorbital sinus injection.” (Page 17, Line 588-589)
(15) “The histological images demonstrated the particle distribution in each organ. The white arrows demonstrate the FITC labeled PEG-Au-BP nanoparticles (green color).” (Page 17, Line 591-592)
(16) “The as-prepared PEG-Au-BP nanoparticles were applied to treat….” (Page 18, Line 605)
(17) “Based on our animal model, PEG-Au-BP nanoparticles could penetrate the BBB,…” (Page 20, Line 721-722)
(18) “In closing, PEG-Au-BP are a potential nanodrug delivery system…” (Page 20, Line 723-724)
- Do the results conclude ??? The authors conclude I guess, they should write in the paper.
Answer:
Thanks for the comment from the reviewer. We have checked the data to be included in the paper and supplementary materials. The “Conclusion” section as shown in the Page 20 (Line 727-738).

Reviewer 3 Report
The manuscript describes the preparation of a drug-conjugated gold nanoparticle system, and the evaluation of its delivery efficiency and anti-cancer capacity against brain cancer.
While the aim of the study is indeed interesting and it represents a huge challenge in cancer research, the presented work lacks the minimum fundamental requirements for a scientific research study aiming to be published in this journal. In particular, the discussion on the chemistry behind the drug-gold conjugate is completely missing.
The nature of the interaction of PEG with gold is not described or discussed, as well as the interaction of the drug with EDC and PEG to crosslink the polymer chains. It is in fact well-known in the literature that thiol-bearing PEGs are required for binding gold surfaces, and no reference to thiols is made in the manuscript. In addition, the drug expose no functional group able to crosslink PEG, and therefore the conjugation approach with a carbodiimide such as EDC is just not convincing.
I am therefore obliged to recommend the Editor for the rejection of the present manuscript.
Author Response
Reviewer 3.
The manuscript describes the preparation of a drug-conjugated gold nanoparticle system, and the evaluation of its delivery efficiency and anti-cancer capacity against brain cancer.
While the aim of the study is indeed interesting and it represents a huge challenge in cancer research, the presented work lacks the minimum fundamental requirements for a scientific research study aiming to be published in this journal. In particular, the discussion on the chemistry behind the drug-gold conjugate is completely missing.
The nature of the interaction of PEG with gold is not described or discussed, as well as the interaction of the drug with EDC and PEG to crosslink the polymer chains. It is in fact well-known in the literature that thiol-bearing PEGs are required for binding gold surfaces, and no reference to thiols is made in the manuscript. In addition, the drug expose no functional group able to crosslink PEG, and therefore the conjugation approach with a carbodiimide such as EDC is just not convincing.
I am therefore obliged to recommend the Editor for the rejection of the present manuscript.
Answer:
Thanks for the comment from the reviewer. We have deleted the description related to the thiol bonding in the “Discussion” section, and the new description was included. Additionally, we have also revised the FTIR spectrum in the Figure 1C.
(1) “The specific peaks of PEG are seen at 3390 cm-1 (-OH stretching), 2880 cm-1 (C-H stretching), 1648 cm-1 (-OH blending), and 1108 cm-1 (C-O-C stretching) [38]. After PEG conjugated with Au, the peak of 3390 cm-1 (-OH bond) shifted to 3397 cm-1, indicating Au nanoparticles were combined with PEG (Figure 1C). Furthermore, the peaks of 1653 cm-1 shifted to 1657 cm-1 (O=C-N, amide bond), showing PEG-Au was cross-linked with BP (Figure 1C).” (Page 7, Line 323-328)
Reference:
38. Loloei, M.; Omidkhah, M.; Moghadassi, A.; Amooghin, A.E. Preparation and characterization of Matrimid® 5218 based binary and ternary mixed matrix membranes for CO2 separation. International Journal of Greenhouse Gas Control 2015, 39, 225-235.
(2) “According to previous literature, the stability of Au nanoparticles could be modified by PEG with the hydrogen bonding, also namely PEGylated Au nanoparticles [50]. PEGylated Au own the ability of anti-immune response as well as have an extended retention time in the bloodstream, while inhibiting the reticuloendothelial system (RES) to uptake [51].” (Page 18-19, Line 633-637)
Reference:
50. Stiufiuc, R.; Iacovita, C.; Nicoara, R.; Stiufiuc, G.; Florea, A.; Achim, M.; Lucaciu, C.M. One-step synthesis of PEGylated gold nanoparticles with tunable surface charge. Journal of Nanomaterials 2013, 2013.
51. Perrault, S.D.; Walkey, C.; Jennings, T.; Fischer, H.C.; Chan, W.C. Mediating tumor targeting efficiency of nanoparticles through design. Nano letters 2009, 9, 1909-1915.
(3) “Furthermore, EDC (1-ethyl-3- dimethylaminopropyl carbodiimide hydrochloride) [53] was used as cross-linking agent to combine BP with PEG-Au nanocarriers. EDC would react with carboxyl group (-COOH) to form hyperactive intermediate. Next, the intermediate would react with primary amine to form Amide bonding [54], which corresponded to the FTIR result (Figure 1C).” (Page 19, Line 640-644)
Reference:
53. Grabarek, Z.; Gergely, J. Zero-length crosslinking procedure with the use of active esters. Analytical biochemistry 1990, 185, 131-135.
54. Nakajima, N.; Ikada, Y. Mechanism of amide formation by carbodiimide for bioconjugation in aqueous media. Bioconjugate chemistry 1995, 6, 123-130.

Reviewer 4 Report
In the present manuscript entitled “Evaluating Delivery Efficiency and Anti-Cancer Capacity of 2 Polyethylene Glycol-Gold Nanoparticles Carried 3 n-butylidenephthalide for Brain Cancer” authors have tested the efficacy of the n-butylidenephthalide against breast cancer. However, before making it into the final publication, certain doughs must be clarified to improve the present manuscript scientifically. Specific comments are as follows;
1. The information provided in Section 2.1 is incomplete. It is not clear which PEG was used for the preparation of PEG-Au-BP. From the information provided it is not clear if this is homobifunctional PEG or heterobifunctional PEG. What kind of bond is being formed between the PEG and Au? Which groups are responsible for stated crosslinking? After crosslinking how authors get PEG-Au solution, I guess authors mean to say PEG-Au nanoparticles! The mixing of 1-ethyl-3-(3-dimethyl aminopropyl)-carbodiimide solution, which is generally known as EDC, with BP is not clear. Generally, EDC is used along with NHS for coupling between the primary amines, or other nucleophiles and carboxylic acids by creating an activated ester leaving group. These groups are absent (or not clear if they are present due to insufficient information) in all the used moieties then how this complex is being formed. It is again not clear how the FITC is being attached to the above-developed nanoparticles. Generally, the complexation is done in molar ratio however the ratio is nowhere presented in the method section. The authors have not described the final BP concentration in the finally synthesized nanoparticles (PEG-Au-BP) then how the dose for all the cell culture studies was decided?
2. As I assume some kind of covalent conjugation (although not clear how this may be happening in the synthesized conjugate) then it would be interesting to investigate the release kinetics of BP from the synthesized conjugate.
3. What is the purpose to use the BABE cell line, as these are not normal brain cell lines? These are Aortic cell lines that are used for hypertension and other heart-related studies.
4. In line 205 “After cell attachment for periods of 24, 48 & 72 hours, 20 μL (0.5 mg/mL) of MTT solution was added” how the authors performed cytotoxicity without adding drugs and formulations.
5. The authors used the same procedure for fluorescence microscopy and flow cytometry in the same section. However, the method of preparation is different in both cases.
6. In section 2.7, cells are incubated in 75% ethanol at -20 o C. Do the cells live at that temperature?
7. In section 2.8, how the treatment is given before attachment of cells, normally it requires at least 12 hours to attach the cells. What is needed to stain cells with DAPI?
8. How the particle size is reduced after conjugating BP (PEG-Au is 110.17 nm PEG-Au-BP is 98.11).
9. The representation of cytotoxicity data is not clear; Cytotoxicity data should be represented in concentration/% cell viability.
10. What is the purpose of satin b-actin and DAPI in all studies?
11. No information is available on which primary and secondary antibodies were used (mouse, rat, goat, etc.)
Author Response
Reviewer 4.
In the present manuscript entitled “Evaluating Delivery Efficiency and Anti-Cancer Capacity of 2 Polyethylene Glycol-Gold Nanoparticles Carried 3 n-butylidenephthalide for Brain Cancer” authors have tested the efficacy of the n-butylidenephthalide against breast cancer. However, before making it into the final publication, certain doughs must be clarified to improve the present manuscript scientifically. Specific comments are as follows;
- The information provided in Section 2.1 is incomplete. It is not clear which PEG was used for the preparation of PEG-Au-BP. From the information provided it is not clear if this is homobifunctional PEG or heterobifunctional PEG. What kind of bond is being formed between the PEG and Au? Which groups are responsible for stated crosslinking? After crosslinking how authors get PEG-Au solution, I guess authors mean to say PEG-Au nanoparticles! The mixing of 1-ethyl-3-(3-dimethyl aminopropyl)-carbodiimide solution, which is generally known as EDC, with BP is not clear. Generally, EDC is used along with NHS for coupling between the primary amines, or other nucleophiles and carboxylic acids by creating an activated ester leaving group. These groups are absent (or not clear if they are present due to insufficient information) in all the used moieties then how this complex is being formed. It is again not clear how the FITC is being attached to the above-developed nanoparticles. Generally, the complexation is done in molar ratio however the ratio is nowhere presented in the method section. The authors have not described the final BP concentration in the finally synthesized nanoparticles (PEG-Au-BP) then how the dose for all the cell culture studies was decided?
Answer:
Thanks for the comment from the reviewer. We have revised the description in the Section 2.1. (Page 3-4, Line 143-156) “PEG solution was purchased from Sigma-Aldrich, USA (average mol wt = 200 kDa). 1 ml of PEG solution (500 µM) was diluted with 24 ml of deionized water to obtain 20 µM PEG solution. Afterwards, 609.6 µl of PEG solution was mixed with 390.4 µl of 50 ppm physical gold nanoparticle solution (Gold NanoTech, Inc, Taiwan) for 2 hours to obtain the PEG-Au solution. n-Butylidenephthalide (BP) solution was purchased from Alfa Aesar, Ward Hill, NY, USA (10 mg/ml, 95%, mol weight: 188.23). To prepare the PEG-Au-BP solution, BP solution was firstly mixed with 1-ethyl-3-(3-dimethyl aminopropyl)-carbodiimide solution (EDC, 15 mM, ThermoFisher, USA) at volume ratio 1:2 to interact for 2 hours at room temperature. 6 µl of mixture was then combined with 250 µl of PEG-Au solution to react at 4oC for 8 hours. The PEG-Au-BP solution with 50 ppm of Au nanoparticles and 78 µg/ml of BP was obtained. Furthermore, to investigate the cellular metabolic process, 0.5 mg/ml of fluorescein isothiocyanate reagent (FITC-conjugated AffiniPure Goat Anti-Rabbit IgG, Jackson ImmunoResearch, USA) was conjugated with PEG-Au-BP through coupling with the amine groups at 4oC for 8 hours in dark at a 50:1 volume ratio.” - As I assume some kind of covalent conjugation (although not clear how this may be happening in the synthesized conjugate) then it would be interesting to investigate the release kinetics of BP from the synthesized conjugate.
Answer:
Thanks for the suggestion from the reviewer. We agree that it would be interesting to investigate the release kinetics of BP from the synthesized conjugate. We will execute the experiments to investigate these nanoparticles’ anti-brain cancer capacity in vivo animal model, and the data will be discussed in the future research. - What is the purpose to use the BABE cell line, as these are not normal brain cell lines? These are Aortic cell lines that are used for hypertension and other heart-related studies.
Answer:
Thanks for the comment from the reviewer. We applied BAEC cell line as non-transformed cells to compared with DBTRG brain cancer cell line. The tumor-cell selective cytotoxicity would be discussed in the present research. (Page 1, Line 35-36) - In line 205 “After cell attachment for periods of 24, 48 & 72 hours, 20 μL (0.5 mg/mL) of MTT solution was added” how the authors performed cytotoxicity without adding drugs and formulations.
Answer:
Thanks for the comment from the Reviewer. We have revised the description in Section 2.4. (Page 5, Line 204-207)
“After incubated overnight for attachment, the cells were cultured with 1 ml various materials containing medium for 24, 48 & 72 hours. Next, the supernatant would be removed after the incubation, and 20 μL (0.5 mg/mL) of MTT solution was added into each well and incubated for 4 hours at 37°C.” - The authors used the same procedure for fluorescence microscopy and flow cytometry in the same section. However, the method of preparation is different in both cases.
Answer:
Thanks for the comment from the reviewer. We have revised the description in the Section 2.5.
(1) “The fluorescence of each sample was observed through a fluorescence microscope (Zeiss Axio Imager A1), and quantified by Image J 5.0 software. Furthermore, the cellular uptake ability was also detected by fluorescein-positive cells through both a flow cytometer and fluorescence-activated Cell Sorting (FACS) software (Becton Dickinson, Canton, MA, USA).” (Page 5, Line 220-224)
(2) “The images were obtained by fluorescence microscope (Zeiss Axio Imager A1) and quantified by Image J 5.0 software. The fluorescein positive cells were also detected FACS software (Becton Dickinson, USA). All experiments were represented in triplicate.” (Page 5, Line 231-234) - In section 2.7, cells are incubated in 75% ethanol at -20 o C. Do the cells live at that temperature?
Answer:
Thanks for the comment from the reviewer. We have revised the description in the Section 2.7. (Page 6, Line 251-254)
“The cells were subsequently collected by a centrifuge (1500 rpm, 3 minutes) and suspended in 75% ethanol (1 mL, −20°C) for 20 minutes. The dead cells were collected and mixed with 500 μL of PBS solution containing 10 μl of PI (1 mg/ml), RNase A (5 mg/ml), and Triton X 100 (0.1%) for 30 minutes.” - In section 2.8, how the treatment is given before attachment of cells, normally it requires at least 12 hours to attach the cells. What is needed to stain cells with DAPI?
Answer:
Thanks for the comment from the reviewer. We have revised the description in the Section 2.8. (Page 6, Line 258-268) “DBTRG cells were firstly seeded at a density of 2´105 cells per well in 6-well culture plates for overnight incubation at 37°C prior to the experiment. Next, the cells were treated with various materials (PEG, PEG-Au, PEG-Au-BP and BP) for 48 hours incubation at 37°C. Afterwards, the DBTRG cells were collected using 0.05% trypsin-EDTA, and treated with anti-Annexin V (green color) and PI (red color) using an Annexin V-FITC/Propidium Iodide (PI) apoptosis detection kit (BD Pharmingen). Test cells were stained with Annexin-V and PI (both 5 μL) and incubated in the dark for 15 minutes at room temperature, following with 400 μL of Annexin-V binding buffer added into each sample. Annexin-V positive cells were considered as apoptotic cells, which were analyzed by BD FACS Calibur flow cytometer (BD Biosciences, Canton, MA, USA). All experiments were represented in triplicate.” - How the particle size is reduced after conjugating BP (PEG-Au is 110.17 nm PEG-Au-BP is 98.11).
Answer:
Thanks for the comment from the reviewer. We have revised the description in the Section 3.1. (Page 7, Line 349-350) and in the Figure 2C (Page 8)
“Additionally, the diameter of Au, PEG-Au, PEG-Au-BP and FITC labeled PEG-Au-BP measured by DLS assay was 42.02±8.95 nm, 177.5±5.2 nm, 246.2±7.0 nm and 288.2±4.2 nm, respectively (Figure 2C).” - The representation of cytotoxicity data is not clear; Cytotoxicity data should be represented in concentration/% cell viability.
Answer:
Thanks for the valuable suggestion from the reviewer. We have revised the presentation of cell viability into percentage in the Figure 3 and the description in the section 3.2. “The cytotoxicity of various nanomaterials was investigated through MTT assay after 24, 48 and 72 hours of incubation. In the non-transformed BAEC cell line (Figure 3A), the cell viability (%) for 24-hour incubation is demonstrated [Control group (BAEC): 100%, PEG group: 89%, PEG-Au group: 95%, PEG-Au-BP group: 43%, and BP group: 36%]. For 48-hour incubation, the results are presented as Control group (BAEC): 100%, PEG group: 120%, PEG-Au group: 101%, PEG-Au-BP group: 59%, and BP group: 91%. Regarding 72-hour incubation, the data revealed Control group (BAEC): 100%, PEG group: 106%, PEG-Au group: 97%, PEG-Au-BP group: 87%, and BP group: 85% (Figure 3A). Furthermore, the cytotoxicity of brain cancer DBTRG cell line, as induced by various materials was examined. The cell viability (%) results at 24, 48 and 72 hours are displayed as follows: 24 hours [Control group (DBTRG): 100%, PEG group: 55%, PEG-Au group: 55%, PEG-Au-BP group: 21%, and BP group: 27%]; 48 hours: [Control group (DBTRG): 100%, PEG group: 85%, PEG-Au group: 84%, PEG-Au-BP group: 33%, and BP group: 37%]; 72 hours: [Control group (DBTRG): 100%, PEG group: 68%, PEG-Au group: 72%, PEG-Au-BP group: 15%, and BP group: 24%] (Figure 3B). The above results demonstrate that PEG-Au-BP could significantly inhibit DBTRG brain cancer cell proliferation when compared to the PEG and PEG-Au groups.” (Page 9, Line 363-379) - What is the purpose of satin b-actin and DAPI in all studies?
Answer:
Thanks for the comment from the reviewer. We have revised the description in the Section 2.5 and the Section 2.9. (1) “Afterwards, the nucleus was stained with 4, 6-diamidino-2-phenylindole (DAPI) nuclear staining (50 μg/mL, Invitrogen, USA) for 10 minutes, and washed twice with PBS.” (Page 5, Line 218-219)
“β-actin antibody was applied to ensure the uniformity of loading.” (Page 6, Line 287-288) - No information is available on which primary and secondary antibodies were used (mouse, rat, goat, etc.)
Answer:
Thanks for the comment from the reviewer. We have revised the description in the Section 2.9 (Page 6, Line 288-290).
“The immunoblots were washed three times with TBST and incubated with HRP-conjugated goat, anti-rabbit or anti-mouse IgG (1:2000 dilution) (Zhongshan Goldenbridge Biotechnology, China) at room temperature for 1 hour.”

Round 2
Reviewer 1 Report
No.
Reviewer 2 Report
The authors addressed the concerns and suggestions of the reviewers.
Reviewer 4 Report
Responses are satisfactory.